# Evaluation of dietary crude protein concentrations, fishmeal, and sorghum inclusions in broiler chickens offered wheat-based diet via Box-Behnken response surface design

Shemil P. Macelline[1,2], Peter V. Chrystal[2,3], Shiva Greenhalgh[1,2], Mehdi Toghyani[1,2], Peter H. Selle[1,4], Sonia Y. Liu[1,2]*

1 Poultry Research Foundation, The University of Sydney, Camden, NSW, Australia, 2 School of Life and Environmental Sciences, Faculty of Science, The University of Sydney, Sydney, NSW, Australia, 3 Complete Feed Solutions, Hornsby, NSW, Australia; Howick, New Zealand, 4 Sydney School of Veterinary Science, The University of Sydney, Sydney, NSW, Australia

* sonia.liu@sydney.edu.au

**Data Availability Statement:** All relevant data are within the paper and its Supporting Information files.

## Abstract

The objective of this study was to investigate the impacts of dietary crude protein (CP), fishmeal and sorghum on nutrient utilisation, digestibility coefficients and disappearance rates of starch and protein, amino acid concentrations in systemic plasma and their relevance to growth performance of broiler chickens using the Box-Behnken response surface design. The design consisted of three factors at three levels including dietary CP (190, 210, 230 g/kg), fishmeal (0, 50, 100 g/kg), and sorghum (0, 150, 300 g/kg). A total of 390 male, off-sex Ross 308 chicks were offered experimental diets from 14 to 35 days post-hatch. Growth performance, nutrient utilisation, starch and protein digestibilities and plasma free amino acids were determined. Dietary CP had a negative linear impact on weight gain where the transition from 230 to 190 g/kg CP increased weight gain by 9.43% (1835 versus 2008 g/bird, P = 0.006). Moreover, dietary CP linearly depressed feed intake (r = -0.486. P < 0.001). Fishmeal inclusions had negative linear impacts on weight gain (r = -0.751, P < 0.001) and feed intake (r = -0.495, P < 0.001). There was an interaction between dietary CP and fishmeal for FCR. However, growth performance was not influenced by dietary inclusions of sorghum. Total plasma amino acid concentrations were negatively related to weight gain (r = -0.519, P < 0.0001). The dietary transition from 0 to 100 g/kg fishmeal increased total amino acid concentrations in systemic plasma by 35% (771 versus 1037 µg/mL, P < 0.001). It may be deduced that optimal weight gain (2157 g/bird), optimal feed intake (3330 g/bird) and minimal FCR (1.544) were found in birds offered 190 g/kg CP diets without fishmeal inclusion, irrespective of sorghum inclusions. Both fishmeal and sorghum inclusions did not alter protein and starch digestion rate in broiler chickens; however, moderate reductions in dietary CP could advantage broiler growth performance.

**Funding:** This study is funded by Australian Research Council Discovery Early Career Researcher Award (DE190101364) and the award supports Dr Sonia Liu's salary and the feeding study. The Australian Government Research and Training Program (RTP) provided scholarship for the PhD candidature of Mr Shemil Macelline. The funders did not have any additional role in the study design, data collection and analysis, decision to publish or preparation of the manuscript.

**Competing interests:** The authors have declared that no competing interests exist.

## Introduction

Developing reduced crude protein (CP) diets where soybean meal partially replaced with supplementary amino acids is a promising nutritional strategy to achieve sustainable chicken meat production with reduced nitrogen excretion and improved bird welfare. Dietary CP reduction from 220 to 160 g/kg decreased nitrogen excretion by 35% and dietary CP reduction from 198 to 169 g/kg reduced foot-pad lesion scores by 59% in broiler chickens as reviewed by Greenhalgh et al. [1]. Moreover, a dietary CP reduction from 222 to 165 g/kg reduced soybean meal inclusions by 74% in Chrystal et al. [2]. However, broiler chickens responded to reduced CP diets inconsistently; for instance, broilers offered maize-based, reduced-CP diets improved growth performance in comparison to standard CP diets; whereas, broilers offered wheat-based, reduced-CP diets displayed inferior growth performance [2].

Starch and protein digestive dynamics in broiler chickens are important for optimising growth performance in broiler chickens offered reduced-CP diets because dietary CP reduction increases non-bound amino acid (NBAA) inclusions and starch content [3]. Both glucose and monomeric amino acids are co-absorbed with $Na^+$ *via* their respective $Na^+$-dependent transport systems; whereas, di- and tri- peptides are absorbed *via* PepT-1 transporter [4, 5]. Glucose and monomeric amino acids may compete for intestinal uptakes [6] and it is possible that such competitions are more pronounced in reduced-CP diets. Liu et al. [7] investigated the influence of diets based on feedstuffs with predetermined starch and protein digestion rate on broiler growth performance from 7 to 35 days post-hatch, where retarding starch digestion rates and/or accelerating protein digestion rates improved FCR and condensed starch:protein digestion rate ratios quadratically (r = 0.648; P < 0.001) improved FCR. In Australia, sorghum is the main alternative feed grain to wheat; starch digestions rates in sorghum are considered slower than wheat under *in vitro* conditions [8]. Moreover, sorghum starch digestion rates were slower than wheat starch by 56% (0.075 versus 0.117 min $^{-1}$) as determined in broiler chickens in Selle et al. [9]. Fishmeal inclusions at 175 g/kg in sorghum-soybean-meal–based diets significantly increased protein disappearance rates in proximal and distal ileum which indicates that fishmeal can be a more rapidly digestible protein source than soybean meal [10]. Previous studies reported the relevance of protein digestion rate in broiler diet using casein [11] and whey protein concentrate [12]. It is intended to apply a more practical feed ingredient in the present study to test the relevance of starch and protein digestive dynamics. Therefore, the objective of the present study was to evaluate the relevance of starch and protein digestive dynamics in reduced CP diets by varying inclusion levels of fishmeal and sorghum in a wheat-soybean-meal based diets.

Box-Behnken design (BBD) is a multivariate optimization design with the advantage of testing multiple nutrients simultaneously with less number of treatments [13]. It was previously used to optimise digestive dynamics and compare relative importance of dietary factors [14, 15]. Therefore, the experimental diets in the present study contained three levels of dietary CP (190, 210, 230 g/kg), three levels of fishmeal (0, 50, 100 g/kg), and three levels of sorghum (0, 150, 300 g/kg) and response surface was plotted to visualise experimental results. The hypothesis was both starch and protein digestive dynamics, which was reflected by variable dietary fishmeal and sorghum inclusions, would influence growth performance and nutrient utilisation; moreover, it was expected the impact of starch and protein digestive dynamics was more pronounced in diets containing lower CP and higher NBAA.

## Materials and methods

This feeding study was conducted in compliance with the guidelines of the Animal Ethics Committee of The University of Sydney (Project number 2019/1516).

## Experimental design

A three factor, three level Box-Behnken response surface experiment with 13 dietary treatments was used to investigate the impact of three dietary CP levels (190, 210, 230 g/kg) with three inclusion levels of fishmeal (0, 50, 100 g/kg) and sorghum (0, 150, 300 g/kg) on growth performance, nutrient utilisation, starch protein digestibility coefficients and plasma amino acid concentrations from 14 to 35 days post-hatch. The factorial arrangement and their respective levels across 13 dietary treatments are shown in Table 1. The central points were 210 g/kg dietary CP level, 50 g/kg fish meal inclusions, and 150 g/kg sorghum inclusions as described for treatment 13M.

## Diet preparation

Dietary composition and nutrient specifications are shown in Tables 2 and 3. The nutritionally equivalent diets were formulated based on near-infrared spectroscopy (NIR) of wheat, sorghum and soybean meal using the AMINONir® Advanced program (Evonik Nutrition & Care GmbH, Hanau, Germany). Experimental diets were based on wheat, soybean meal, canola meal with or without sorghum and fishmeal. Varying levels of maize starch ranged from 7.50 to 150 g/kg were added to Diets 1A, 3C, 6F, 8H, 10J, and 12L in order to make all diets iso-energetic (13.0 MJ/kg). Experimental diets were formulated based on digestible amino acids with 10 g/kg of lysine across all dietary treatments. NBAA including lysine, methionine, threonine, tryptophan, valine, arginine, isoleucine, leucine, and glycine were supplemented to maintain similar ideal protein ratios for TSAA, Thr, Trp, Ile, Val, Arg and Gly-equivalent. A commercial starter diet based on wheat and soybean meal with 12.13 MJ/kg energy and 220 g/kg crude protein, was offered to broiler chickens from 1 to 13 days post-hatch. Acid insoluble ash (AIA; Celite™ World Minerals, Lompoc, CA, USA) was included at 20 g/kg in diets as an inert marker in order to determine nutrient digestibility coefficients in two small intestinal sites. Sorghum and wheat were mediumly ground (4.0 mm hammer-mill screen) prior to being blended into the complete diets. All diets were cold-pelleted and contained xylanase and offered to broiler chickens from 14 to 35 days post-hatch.

## Bird management

A total of 390 off-sex 14-days old male broilers (Ross 308) were randomly distributed into 65 battery cages each of 6 birds (13 treatments × 5 replicates). The variance of average initial body

**Table 1. Factor levels for Box-Behnken design in present study.**

| Diet | Sorghum (g/kg) | Fishmeal (g/kg) | Dietary crude protein (g/kg) |
|------|----------------|-----------------|------------------------------|
| 1A | 300 | 100 | 210 |
| 2B | 300 | 0 | 210 |
| 3C | 0 | 100 | 210 |
| 4D | 0 | 0 | 210 |
| 5E | 150 | 100 | 230 |
| 6F | 150 | 100 | 190 |
| 7G | 150 | 0 | 230 |
| 8H | 150 | 0 | 190 |
| 9I | 300 | 50 | 230 |
| 10J | 300 | 50 | 190 |
| 11K | 0 | 50 | 230 |
| 12L | 0 | 50 | 190 |
| 13M | 150 | 50 | 210 |

**Table 2. Composition of experimental diets.**

| Ingredients (g/kg) | Experimental diets | | | | | | | | | | | | |
|---|---|---|---|---|---|---|---|---|---|---|---|---|---|
| | 1A | 2B | 3C | 4D | 5E | 6F | 7G | 8H | 9I | 10J | 11K | 12L | 13M |
| Wheat | 392 | 301 | 692 | 608 | 492 | 328 | 392 | 525 | 272 | 276 | 578 | 569 | 502 |
| Sorghum | 300 | 300 | - | - | 150 | 150 | 150 | 150 | 300 | 300 | - | - | 150 |
| Maize starch | 8.69 | - | 14.9 | - | - | 150 | - | 7.50 | - | 117 | - | 128 | - |
| Soybean meal | 36.1 | 172 | 30.7 | 163 | 119 | 20.1 | 253 | 91.5 | 171 | 59.1 | 163 | 52.4 | 92.9 |
| Canola meal | 83.8 | 100 | 81.7 | 100 | 56.8 | 100 | 72.4 | 100 | 100 | 100 | 100 | 100 | 100 |
| Fishmeal | 100 | - | 100 | - | 100 | 100 | - | - | 50 | 50 | 50 | 50 | 50 |
| Soybean oil | 33.6 | 62.0 | 36.0 | 64.7 | 42.2 | 52.9 | 70.0 | 51.8 | 58.8 | 35.2 | 61.8 | 37.1 | 50.2 |
| l-lysine HCl | 2.47 | 3.08 | 2.48 | 3.13 | 0.48 | 3.30 | 1.19 | 5.27 | 0.73 | 4.28 | 0.76 | 4.32 | 2.82 |
| d,l-methionine | 1.52 | 2.05 | 1.25 | 1.76 | 1.00 | 2.25 | 1.58 | 2.43 | 1.27 | 2.48 | 0.97 | 2.23 | 1.59 |
| l-threonine | 1.13 | 1.06 | 1.23 | 1.16 | 0.33 | 1.84 | 0.32 | 2.06 | 0.20 | 1.91 | 0.29 | 2.02 | 1.14 |
| l-tryptophan | - | - | 0.02 | - | - | 0.28 | - | 0.11 | - | 0.19 | - | 0.22 | - |
| l-valine | 0.09 | 0.14 | 0.27 | 0.32 | - | 1.24 | - | 1.37 | - | 1.23 | - | 1.43 | 0.19 |
| l-arginine | 2.66 | 5.55 | 2.35 | 5.16 | 3.17 | 3.23 | 6.11 | 5.12 | 4.22 | 4.36 | 3.83 | 4.04 | 3.75 |
| l-isoleucine | 0.25 | 0.11 | 0.39 | 0.26 | - | 1.27 | - | 1.31 | - | 1.23 | - | 1.40 | 0.27 |
| l-leucine | - | - | - | - | - | - | - | - | - | - | - | 0.79 | - |
| Glycine | 4.33 | 0.47 | 3.90 | 0.04 | 2.48 | 5.88 | - | 2.03 | 0.74 | 4.19 | 0.28 | 3.80 | 2.18 |
| Salt | - | 0.79 | - | 0.77 | 0.73 | - | 1.51 | - | 1.14 | - | 1.12 | - | 0.37 |
| NaHCO$_3$ | 4.77 | 4.90 | 4.68 | 4.82 | 3.80 | 4.78 | 3.96 | 5.99 | 3.73 | 5.44 | 3.64 | 5.34 | 4.78 |
| Limestone | 4.94 | 1.74 | 5.04 | 11.8 | 4.89 | 4.78 | 11.7 | 12.1 | 8.01 | 8.39 | 8.10 | 8.47 | 8.34 |
| MDCP[2] | - | 12.4 | - | 12.3 | - | 0.36 | 12.5 | 13.0 | 5.44 | 6.71 | 5.38 | 6.68 | 6.00 |
| Xylanase | 0.10 | 0.10 | 0.10 | 0.10 | 0.10 | 0.10 | 0.10 | 0.10 | 0.10 | 0.10 | 0.10 | 0.10 | 0.10 |
| Choline chloride | 0.90 | 0.90 | 0.90 | 0.90 | 0.90 | 0.90 | 0.90 | 0.90 | 0.90 | 0.90 | 0.90 | 0.90 | 0.90 |
| Celites | 20.0 | 20.0 | 20.0 | 20.0 | 20.0 | 20.0 | 20.0 | 20.0 | 20.0 | 20.0 | 20.0 | 20.0 | 20.0 |
| Sand | - | - | - | - | - | 46.8 | - | - | - | - | - | - | - |
| Vit min premix[1] | 2.00 | 2.00 | 2.00 | 2.00 | 2.00 | 2.00 | 2.00 | 2.00 | 2.00 | 2.00 | 2.00 | 2.00 | 2.00 |

[1]Vitamin-trace mineral premix supplied per tonne of feed; [million international units, MIU] retinol 12, cholecalciferol 5, [g] tocopherol 50, menadione3, thiamine 3, riboflavin 9, pyridoxine 5, cobalamin 0.025, niacin 50, pantothenate 18, folate 2, biotin 0.2, copper 20, iron 40 manganese 110, cobalt 0.25, iodine 1, molybdenum 2, zinc 90, selenium 0.3.

[2]MDCP, Mono Dicalcium Phosphate.

weight was maintained at 1.02% between cages. The dimensions of the cages were 750 mm in width and depth and 500 mm in height. An environmentally controlled housing facility was used and birds had *ad-libitum* access to feed and water. An initial room temperature of 32 ± 1˚C was maintained for the first week, which was gradually decreased to 22 ± 1˚C by the end of the third week and maintained at this temperature with a '18-hours-on-6-hours-off' lighting regime for the duration of the feeding study. Initial and final body weights were recorded to determine weight gain. FCR was calculated from feed intake divided by weight gain for the experiment period and any culled/dead bird's body weights were recorded to adjust feed intakes and FCR calculations.

## Sample collection and chemical analysis

Total excreta were collected from 27 to 29 days post-hatch and feed intake during this period was measured separately to determine apparent metabolizable energy (AME), metabolisable energy to gross energy ratio (ME:GE), N retention and N-corrected apparent metabolisable energy (AMEn). Excreta were dried in a forced-air oven at 80˚C for 24 hours and the gross

**Table 3. Nutrient specification of experimental diets.**

| Nutrient (g/kg) | Diet 1A | Diet 2B | Diet 3C | Diet 4D | Diet 5E | Diet 6F | Diet 7G | Diet 8H | Diet 9I | Diet 10J | Diet 11K | Diet 12L | Diet 13M |
|---|---|---|---|---|---|---|---|---|---|---|---|---|---|
| **Calculated values** | | | | | | | | | | | | | |
| ME (MJ/kg) | 13.0 | 13.0 | 13.0 | 13.0 | 13.0 | 13.0 | 13.0 | 13.0 | 13.0 | 13.0 | 13.0 | 13.0 | 13.0 |
| Crude protein | 210 | 210 | 210 | 210 | 230 | 190 | 230 | 190 | 230 | 190 | 230 | 190 | 210 |
| Starch | 440 | 377 | 440 | 377 | 400 | 428 | 340 | 427 | 360 | 462 | 358 | 463 | 406 |
| Starch: protein | 2.10 | 1.80 | 2.10 | 1.80 | 1.74 | 2.25 | 1.48 | 2.25 | 1.57 | 2.43 | 1.57 | 2.44 | 1.93 |
| Calcium | 8.25 | 8.25 | 8.25 | 8.25 | 8.25 | 8.25 | 8.25 | 8.25 | 8.25 | 8.25 | 8.25 | 8.25 | 8.25 |
| Phosphorous available | 4.13 | 4.13 | 4.13 | 4.13 | 4.13 | 4.13 | 4.13 | 4.13 | 4.13 | 4.13 | 4.13 | 4.13 | 4.13 |
| Crude fat | 67.2 | 87.2 | 61.9 | 82.3 | 71.4 | 75.9 | 89.7 | 73.7 | 88.2 | 64.0 | 83.6 | 57.0 | 76.1 |
| Crude fibre | 22.5 | 25.7 | 21.3 | 24.6 | 20.8 | 20.4 | 23.2 | 24.2 | 25.8 | 23.1 | 24.6 | 21.5 | 24.3 |
| *Digestible amino acids* | | | | | | | | | | | | | |
| Lysine | 10.0 | 10.0 | 10.0 | 10.0 | 10.0 | 10.0 | 10.0 | 10.0 | 10.0 | 10.0 | 10.0 | 10.0 | 10.0 |
| TSAA | 7.40 | 7.40 | 7.40 | 7.40 | 7.40 | 7.40 | 7.40 | 7.40 | 7.40 | 7.40 | 7.40 | 7.40 | 7.40 |
| Threonine | 6.70 | 6.70 | 6.70 | 6.70 | 6.70 | 6.70 | 6.70 | 6.70 | 6.70 | 6.70 | 6.70 | 6.70 | 6.70 |
| Tryptophan | 1.90 | 2.14 | 1.90 | 2.11 | 2.20 | 1.90 | 2.42 | 1.90 | 2.34 | 1.90 | 2.32 | 1.90 | 2.01 |
| Isoleucine | 7.00 | 7.00 | 7.00 | 7.00 | 7.75 | 7.00 | 7.82 | 7.00 | 7.85 | 7.00 | 7.72 | 7.00 | 7.00 |
| Leucine | 13.2 | 13.5 | 11.6 | 11.9 | 14.1 | 10.7 | 14.3 | 10.8 | 15.0 | 11.6 | 13.5 | 10.7 | 12.5 |
| Arginine | 10.4 | 10.4 | 10.4 | 10.4 | 10.4 | 10.4 | 10.4 | 10.4 | 10.4 | 10.4 | 10.4 | 10.4 | 10.4 |
| Valine | 8.00 | 8.00 | 8.00 | 8.00 | 8.00 | 8.84 | 8.71 | 8.00 | 8.95 | 8.00 | 8.79 | 8.00 | 8.00 |
| Histidine | 3.92 | 4.13 | 3.97 | 4.17 | 4.52 | 3.41 | 4.68 | 3.51 | 4.63 | 3.43 | 4.68 | 3.46 | 4.05 |
| Phenylalanine | 5.53 | 8.13 | 5.35 | 7.95 | 6.66 | 4.25 | 9.19 | 6.80 | 7.96 | 5.61 | 7.79 | 5.39 | 6.72 |
| Tyrosine | 5.43 | 5.19 | 5.29 | 5.03 | 6.39 | 4.68 | 6.11 | 4.10 | 6.23 | 4.46 | 6.09 | 4.29 | 5.19 |
| Proline | 9.78 | 11.8 | 10.8 | 12.9 | 11.1 | 7.58 | 13.0 | 11.6 | 11.5 | 9.08 | 12.5 | 10.0 | 11.4 |
| Aspartic acid | 3.63 | 9.26 | 4.92 | 10.4 | 7.88 | 2.58 | 1.34 | 6.81 | 9.08 | 4.07 | 10.3 | 5.27 | 6.75 |
| Glutamic acid | 17.4 | 24.0 | 28.2 | 34.9 | 27.3 | 13.8 | 33.5 | 26.5 | 22.8 | 14.7 | 33.7 | 25.2 | 25.7 |
| Glycine | 8.57 | 6.83 | 8.51 | 6.78 | 7.74 | 9.38 | 7.29 | 7.62 | 6.95 | 8.53 | 6.89 | 8.51 | 7.68 |
| Serine | 5.23 | 7.66 | 5.30 | 7.73 | 6.39 | 4.08 | 8.74 | 6.55 | 7.49 | 5.27 | 7.57 | 5.31 | 6.46 |
| Gly- equivalent | 12.3 | 12.3 | 12.3 | 12.3 | 12.3 | 12.3 | 12.3 | 12.3 | 12.3 | 12.3 | 12.3 | 12.3 | 12.3 |
| Alanine | 1.97 | 3.98 | 2.91 | 4.88 | 3.73 | 1.47 | 5.69 | 3.38 | 3.86 | 1.96 | 4.78 | 2.85 | 3.32 |
| Sodium | 1.90 | 1.90 | 1.90 | 1.90 | 1.90 | 1.90 | 1.90 | 1.90 | 1.90 | 1.90 | 1.90 | 1.90 | 1.90 |
| Potassium | 4.98 | 7.33 | 4.63 | 6.95 | 6.32 | 4.14 | 8.61 | 5.59 | 7.59 | 5.06 | 7.22 | 4.69 | 5.92 |
| Chloride | 1.80 | 1.80 | 1.80 | 1.80 | 1.80 | 1.80 | 1.80 | 1.80 | 1.80 | 1.80 | 1.80 | 1.80 | 1.80 |
| DEB | 159 | 219 | 150 | 209 | 193 | 135 | 251 | 174 | 226 | 159 | 216 | 150 | 183 |
| **Analysed values** | | | | | | | | | | | | | |
| GE (MJ/kg) | 16.5 | 17.1 | 16.6 | 17.1 | 16.7 | 16.0 | 17.3 | 16.8 | 17.1 | 16.5 | 17.2 | 16.5 | 17.0 |
| Crude protein | 192 | 201 | 195 | 202 | 219 | 179 | 222 | 183 | 216 | 182 | 223 | 184 | 197 |
| Starch | 392 | 351 | 405 | 343 | 369 | 386 | 297 | 360 | 359 | 422 | 322 | 429 | 400 |
| Starch: protein | 2.04 | 1.75 | 2.08 | 1.70 | 1.68 | 2.16 | 1.34 | 1.97 | 1.66 | 2.32 | 1.44 | 2.33 | 2.03 |
| Total NBAA | 12.5 | 12.5 | 11.9 | 11.8 | 7.47 | 19.3 | 9.20 | 19.7 | 7.15 | 19.9 | 6.13 | 20.3 | 11.9 |

energy **(GE)** of excreta and diets were determined using an adiabatic bomb calorimeter (Parr 1281 bomb calorimeter, Parr Instruments Co., Moline, IL, USA). The AME values (MJ/kg) of the diets were calculated on a dry matter basis from the following equation:

$$AME_{Diet} = \frac{(\text{Feed intake} \times GE_{Diet}) - (\text{Excreta output} \times GE_{Excreta})}{(\text{Feed intake})}$$

N contents of diets and excreta were determined using a nitrogen determinator (Leco

Corporation, St Joseph, MI) and N retentions calculated from the following equation:

$$\text{Retention (\%)} = \frac{(\text{Feed intake} \times \text{Nutrient}_{\text{diet}}) - (\text{Excreta output} \times \text{Nutrient}_{\text{excreta}})}{(\text{Feed intake} \times \text{Nutrient}_{\text{diet}})} \times 100$$

N-corrected AME (AMEn MJ/kg DM) values were calculated by correcting N retention to zero using the factor of 36.54 kJ/g N retained in the body [16].

At day 34, three birds at random were selected from each cage and blood samples were taken from the brachial vein to determine the concentrations of 20 amino acids in systemic plasma. Collected blood samples were then centrifuged and the decanted plasma samples were then kept at −80˚C before analysis. Amino acids concentration in systemic plasma was quantified by 6-aminoquinolyl-N-hydroxysuccinimidyl carbamate (AQC; Waters™ AccQTag Ultra; www.waters.com) followed by separation of the derivatives and quantification by reversed phase ultra-performance liquid chromatography [17]. All amino acids were detected by UV absorbance.

On day 35, all birds were euthanized by intravenous injection of sodium pentobarbitone and digesta samples were collected in their entirety from the distal jejunum and distal ileum to determine apparent digestibility coefficients protein (N) and starch in the distal jejunum and ileum. Small intestines were removed from euthanised birds and samples of digesta were gently expressed from the distal jejunum (below the mid-point between duodenal loop and Meckel's diverticulum) and distal ileum (below the mid-point between Meckel's diverticulum and the ileo-caecal junction) in their entirety and pooled for each cage. The digesta samples were then freeze-dried. Starch concentrations in diets and digesta were determined by a procedure based on dimethyl sulphoxide, α-amylase and amyloglucosidase as described by Mahasukhonthachat et al. [18]. N concentrations were determined as already stated and AIA concentrations were determined by the method of Siriwan et al. [19]. The apparent digestibility coefficients for starch and protein (N) in two small intestinal sites were calculated from the following equation:

$$\text{Digestibility Coefficient} = \frac{(\text{Nutrient}/\text{AIA})_{\text{diet}} - (\text{Nutrient}/\text{AIA})_{\text{digesta}}}{(\text{Nutrient}/\text{AIA})_{\text{diet}}}$$

Starch and protein (N) disappearance rates (g/bird/day) were deduced from the following equation:

*Dissaperence rate = Feed intake × Dietary nutrient concentration × digestibility coefficient*

## Statistical analyses

The experimental units were replicate cage means (6 birds per cage) and statistical procedures included model prediction and linear regression analysis. The surface plots of 3-factor, 3-level Box-Behnken design were obtained by R 3.5.3 software. Best fitted models for response surface designs were predicted by combinations of first- and second-degree polynomial regressions. In model prediction, non-significant coefficients were omitted and the equations were re-predicted with only significant coefficients for each response variable. When more than one models were fitted and significantly different, "Akaike Information Criterion" was used for model comparison and selection. Solver function in Microsoft excel was used to calculate best dietary CP levels with optimal inclusions of fishmeal and sorghum based on response surface design equations.

## Results

The influence of dietary treatments on weight gain, feed intake, FCR, and relative fat pad weights are shown in Table 4. The overall average weight gain and feed intake for all treatments from 14 to 35 days post-hatch were higher than 2019 Ross performance objectives by 3.41% (1912 versus 1849 g/bird) and 4.48% (3052 versus 2921 g/bird), respectively and FCR was comparable (1.601 versus 1.579). The mortality rate in the present study was 1.28% which was not influenced by the dietary treatments (P = 0.77).

The response surface and contour plot for weight gain is illustrated in Fig 1 and there was no interaction between dietary factors. However, increasing dietary CP and fishmeal inclusions linearly reduced the weight gain regardless of the sorghum inclusion. According to the coefficient of the equation, the negative impact of dietary CP was greater than fishmeal inclusion, as described by the following equation,

$$Weight\ Gain\ (g) = 2982 - 4.3425\ CP - 3.1510\ FM,\ (R^2 = 0.736,\ P < 0.001).$$

Based on the above equation, the optimal weight gain of 2157 g/bird was predicted with 190 g/kg dietary CP and no fishmeal regardless of sorghum inclusions.

**Table 4. Effect of dietary treatments on growth performance, and relative abdominal fat-pad weights in broiler chickens from 14 to 35 days post-hatch.**

| Diet | Weight gain (g/bird) | Feed intake (g/bird) | FCR (g/g) | Relative fat-pad weights (g/kg) |
|---|---|---|---|---|
| 1A | 1764 | 2883 | 1.642 | 13.3 |
| 2B | 2053 | 3160 | 1.540 | 12.7 |
| 3C | 1682 | 2783 | 1.655 | 12.7 |
| 4D | 2063 | 3144 | 1.525 | 9.74 |
| 5E | 1678 | 2889 | 1.723 | 12.7 |
| 6F | 1875 | 3158 | 1.685 | 16.0 |
| 7G | 2002 | 3128 | 1.563 | 10.3 |
| 8H | 2140 | 3270 | 1.529 | 13.0 |
| 9I | 1800 | 2934 | 1.630 | 11.4 |
| 10J | 1953 | 3129 | 1.602 | 15.8 |
| 11K | 1857 | 2901 | 1.561 | 11.0 |
| 12L | 2064 | 3267 | 1.583 | 14.6 |
| 13M | 1925 | 3024 | 1.571 | 12.9 |
| **Crude protein (g/kg)** | | | | |
| 190 | 2008 | 3206 | 1.600 | 14.9 |
| 210 | 1897 | 2999 | 1.587 | 12.3 |
| 230 | 1835 | 2963 | 1.619 | 11.3 |
| Linear relationship (r =) | -0.414 | -0.486 | 0.094 | -0.620 |
| P-value | 0.006 | <0.0001 | 0.457 | <0.0001 |
| **Sorghum (g/kg)** | | | | |
| 0 | 1917 | 3024 | 1.581 | 12.0 |
| 150 | 1924 | 3094 | 1.614 | 13.0 |
| 300 | 1893 | 3027 | 1.603 | 13.3 |
| Linear relationship (r =) | -0.058 | 0.005 | 0.108 | 0.228 |
| P-value | 0.648 | 0.967 | 0.393 | 0.068 |
| **Fishmeal (g/kg)** | | | | |
| 0 | 2065 | 3176 | 1.539 | 11.4 |
| 50 | 1920 | 3051 | 1.589 | 13.2 |
| 100 | 1750 | 2928 | 1.676 | 13.7 |
| Linear relationship (r =) | -0.751 | -0.495 | 0.660 | 0.395 |
| P-value | <0.0001 | <0.0001 | <0.001 | 0.0011 |

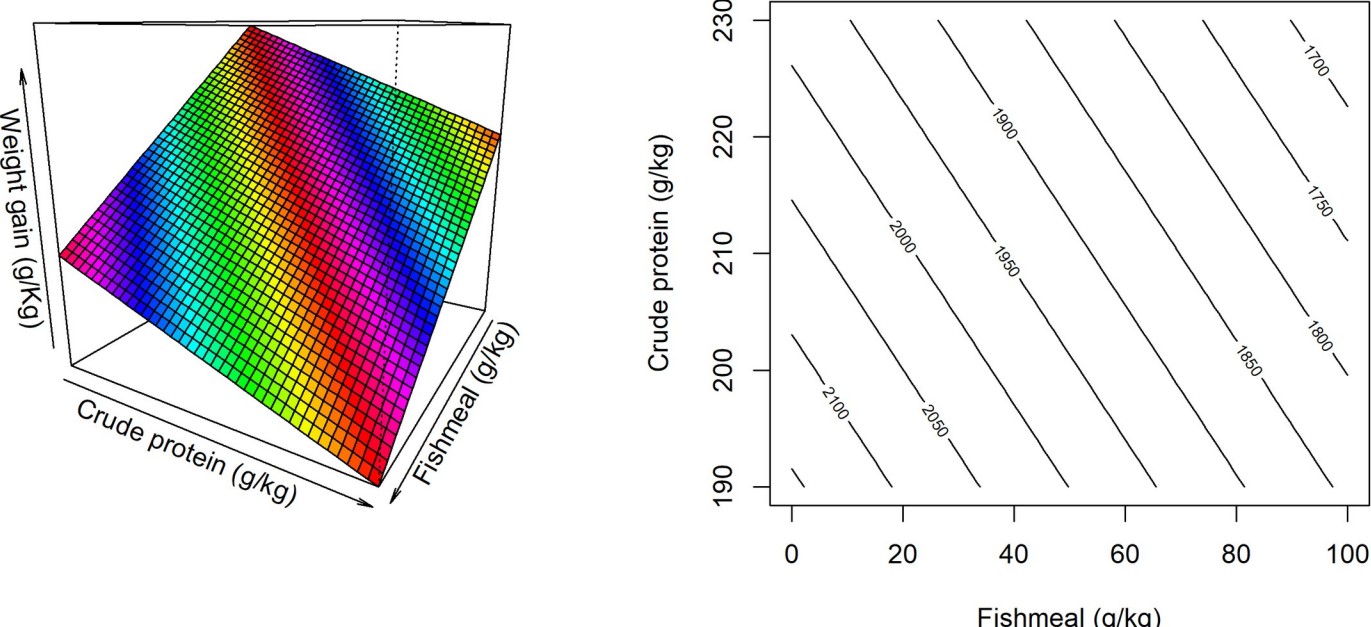

**Fig 1. Response surface and contour plot describing influence of fishmeal inclusion and dietary crude protein level on weight gain in broiler chickens from 14–35 days post-hatch.**

The response surface for feed intake is illustrated in Fig 2. Only fishmeal inclusion and dietary CP level influenced feed intake where increasing fishmeal inclusion depressed feed intake and dietary CP had quadratic relationship with feed intake. The predicted equation for the

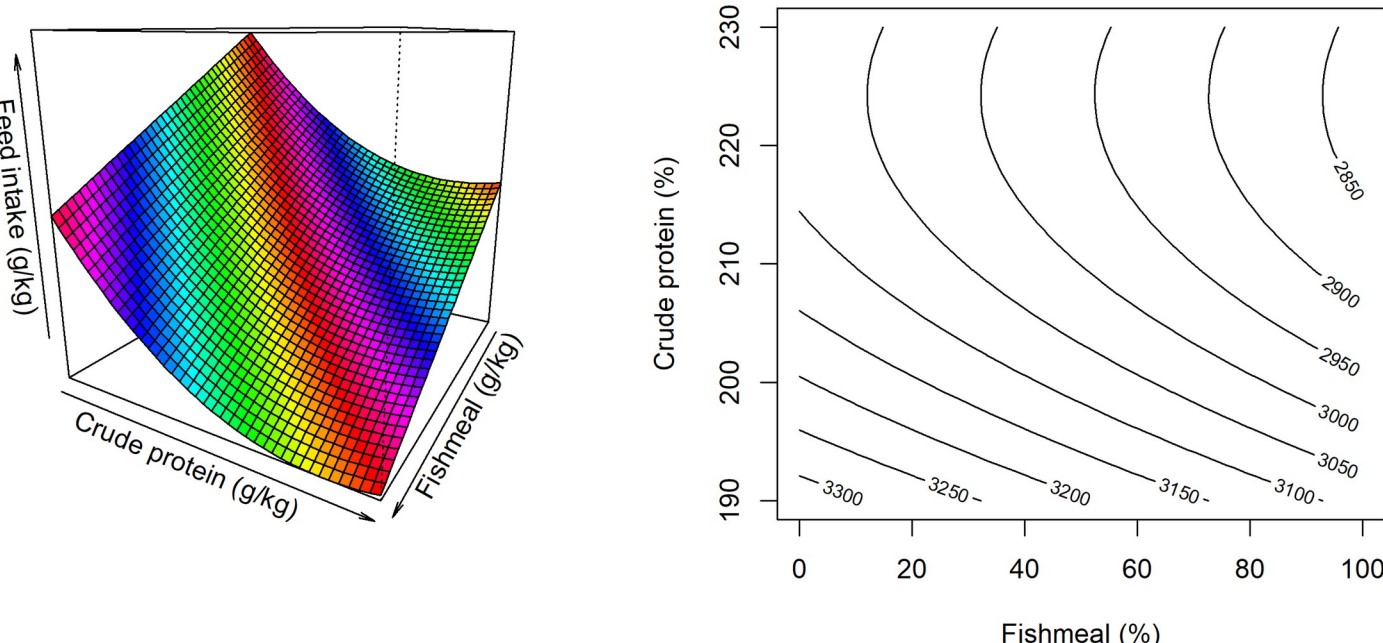

**Fig 2. Response surface and contour plot describing influence of fishmeal inclusion and dietary crude protein level on feed intake in broiler chickens from 14–35 days post-hatch.**

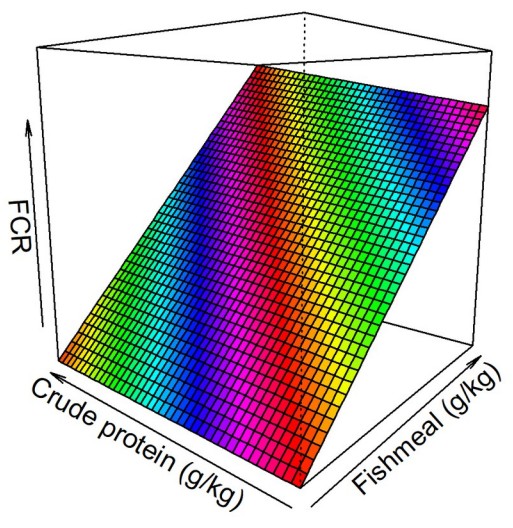
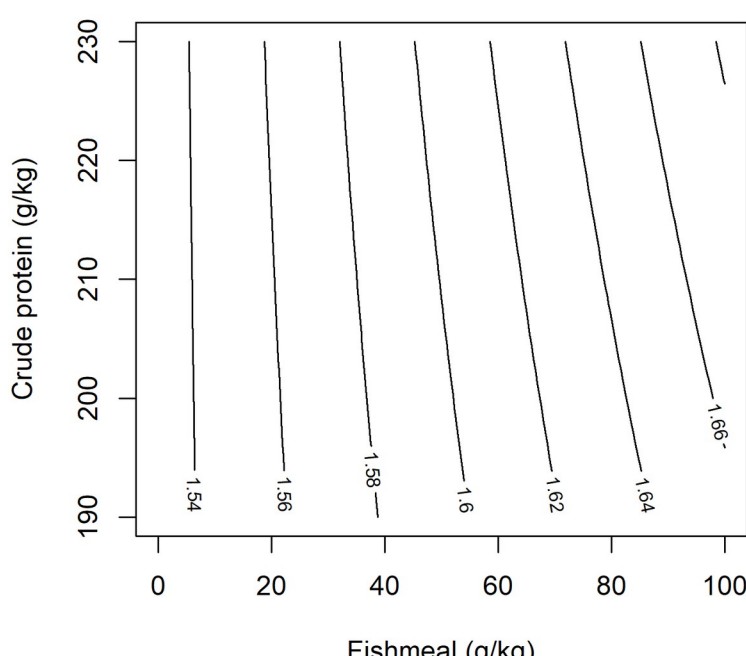

**Fig 3. Response surface and contour plot describing influence of fishmeal inclusion and dietary crude protein level on FCR in broiler chickens from 14–35 days post-hatch.**

feed intake is,

$$\text{Feed intake (g)} = 13806 - 2.472 \times FM - 95.68 \times CP + 2.133 \times CP^2, \ (R^2 = 0.503, \ P < 0.001).$$

The response surface and contour plot for FCR is illustrated in Fig 3. There was a CP and fishmeal interaction on FCR. That negative impact of fish meal inclusions on FCR was more pronounced in diets containing 230 g/kg CP than diets containing 190 g/kg CP. The response of FCR was described by the following equation,

$$FCR = 1.532 + 6.54 \times 10^{-6} \times CP \times FM, \ (R^2 = 0.434, \ P < 0.001).$$

Fig 4 illustrates response surface and contour plots describing the relationship between fish meal inclusions, sorghum inclusions and dietary CP with relative fat pad weights. The surface design shows that increasing fish meal and sorghum inclusions increased the relative fat pad weights whereas dietary CP had negative impact. The predicted equation for relative fat pad weight is as follows,

*Relative fat pad weight* (g)
$$= 28.8552 + 0.03418 \times FM - 0.08776 \times CP + 0.00824 \times Sorghum - 7.9 \times 10^{-5} \ FM \times Sorghum, \ (R^2 = 0.614, \ P < 0.001).$$

The predicted lowest relative fat pad weight of 8.67 g/kg was obtained at 230 g/kg dietary CP with no inclusion of fish meal and sorghum.

The impact of dietary treatments on nutrient utilization including AME, ME:GE, nitrogen retention and N corrected apparent metabolisable energy in broiler chickens from 27–29 post-hatch is reported in Table 5. The response surface and contour plots for ME:GE ratio is

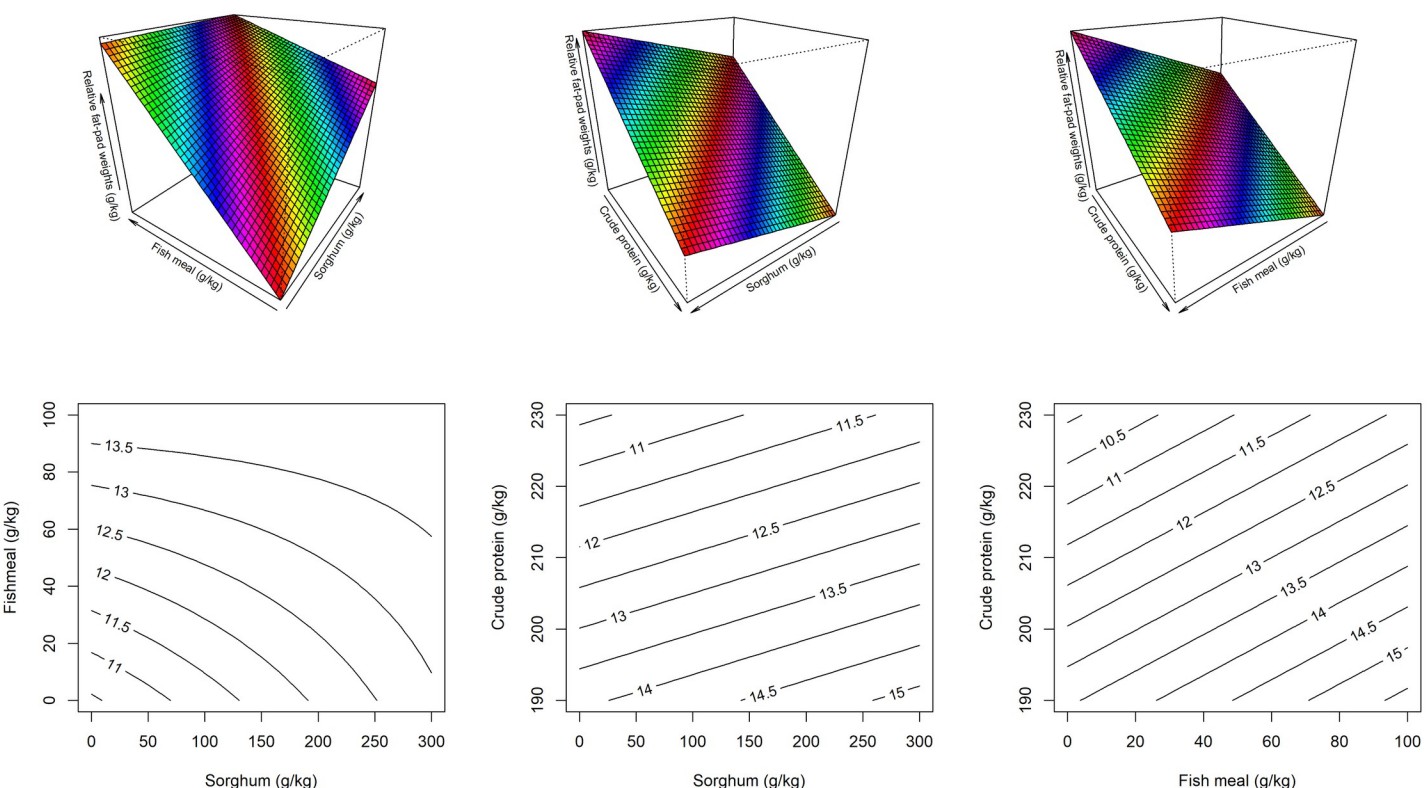

**Fig 4. The response surface designs and contour plots describing the impact of dietary crude protein level together with sorghum and fishmeal inclusions on relative fat pad weights in broiler chickens at 35 days post-hatch (left–slice at 210 g/kg of CP; middle–slice at 5 g/kg of FM; right–slice at 150 g/kg of sorghum).**

illustrated in Fig 5. Increasing fishmeal and sorghum inclusion and decreasing dietary CP resulted higher ME:GE ratio. There is an interaction between fishmeal and CP on ME:GE ratio where higher inclusion of fishmeal to reduced CP diets increased ME:GE to larger extent than under high CP diets (Fig 5 right). The predicted equation for response surface design for ME: GE ratio as follows,

$$ME : GE = 0.86 + 2.625 \times 10^{-3} \times FM + 3.50 \times 10^{-5} \times Sorghum - 4.38 \times 10^{-4} \times CP - 1.05 \times 10^{-5} \times FM \times CP, \ (R^2 = 0.711, \ P < 0.0001)$$

The highest predicted ME:GE ratio of 0.85 was estimated at 190 g/kg of dietary CP level without fish meal and sorghum inclusions.

The effect of dietary treatments on apparent digestibility coefficients and disappearance rates of starch in distal jejunum and distal ileum at 35 days post-hatch is reported in Table 6. Average starch digestibility coefficients in distal jejunum and distal ileum are 0.966 and 0.995 respectively. Crude protein concentrations linearly decreased starch disappearance rate in the distal jejunum (r = -0.800, P < 0.001) and distal ileum (r = -0.812, P < 0.001). Sorghum inclusions had no impact on starch digestibility and disappearance rate. Fishmeal inclusions slightly increased starch digestibility in distal ileum from 0.995 to 0.998 (r = 0.334, P = 0.009). The

**Table 5. Effect of dietary treatments on nutrient utilisation (AME: Apparent metabolisable energy, ME:GE: Metabolisable to gross energy ratio, N: Nitrogen retention, AMEn: N-corrected AME) in broiler chicken from 27 to 29 days post-hatch.**

| Diet | AME (MJ/kg) | ME:GE ratio | N retention (%) | AMEn (MJ/kg) |
|---|---|---|---|---|
| 1A | 15.49 | 0.82 | 65.52 | 14.38 |
| 2B | 15.46 | 0.79 | 66.81 | 14.14 |
| 3C | 15.53 | 0.82 | 65.93 | 14.45 |
| 4D | 15.04 | 0.77 | 61.78 | 13.81 |
| 5E | 15.26 | 0.80 | 63.73 | 14.04 |
| 6F | 15.45 | 0.85 | 71.58 | 14.22 |
| 7G | 15.32 | 0.77 | 62.21 | 13.99 |
| 8H | 15.13 | 0.78 | 7076 | 13.86 |
| 9I | 15.23 | 0.77 | 60.93 | 14.05 |
| 10J | 15.60 | 0.83 | 69.62 | 14.44 |
| 11K | 14.99 | 0.77 | 61.09 | 13.79 |
| 12L | 15.20 | 0.81 | 68.78 | 13.92 |
| 13M | 15.35 | 0.80 | 64.10 | 14.21 |
| **Crude protein (g/kg)** | | | | |
| 190 | 15.34 | 0.82 | 70.18 | 13.97 |
| 210 | 15.37 | 0.80 | 64.83 | 14.20 |
| 230 | 15.20 | 0.78 | 61.99 | 13.96 |
| Linear relationship (r =) | -0.185 | -0.554 | -0.709 | -0.160 |
| P-value | 0.140 | <0.0001 | <0.0001 | 0.203 |
| **Sorghum (g/kg)** | | | | |
| 0 | 15.19 | 0.79 | 64.40 | 13.99 |
| 150 | 15.30 | 0.80 | 66.48 | 14.06 |
| 300 | 15.44 | 0.80 | 65.72 | 14.26 |
| Linear relationship (r =) | 0.319 | 0.126 | 0.115 | 0.303 |
| P-value | 0.009 | 0.310 | 0.364 | 0.014 |
| **Fishmeal (g/kg)** | | | | |
| 0 | 15.23 | 0.78 | 65.39 | 13.95 |
| 50 | 15.27 | 0.82 | 64.90 | 14.08 |
| 100 | 15.43 | 0.80 | 66.69 | 14.27 |
| Linear relationship (r =) | 0.247 | 0.598 | 0.113 | 0.368 |
| P-value | 0.047 | <0.001 | 0.372 | 0.003 |

results of Apparent digestibility coefficients, disappearance rates of protein and starch: protein disappearance rate ratios in distal jejunum and distal ileum are shown in Table 7. Dietary crude protein did not influence apparent protein digestibility coefficients in the distal jejunum and ileum; but it linearly reduced starch and protein disappearance rate ratios in the distal jejunum (r = -0.639, P < 0.001) and ileum (r = -0.779, P < 0.001). Sorghum inclusions had no impact on apparent jejunal protein digestibilities but increasing sorghum inclusion linearly decreased ileal protein digestibility (r = -0.260, P = 0.036). Fishmeal inclusion also had no impact on apparent protein digestibility coefficients in the distal jejunum and ileum. However, increasing fishmeal inclusion increased starch and protein disappearance rate ratios in the distal jejunum (r = 0.342, P = 0.005) and distal ileum (r = 0.375, P = 0.005).

The impact of dietary treatments on free amino acids concentrations in the systemic plasma is shown in Tables 8 and 9. Dietary crude protein linearly increased plasma concentrations of Arg, His, Ile, Leu, Phe, Asn+Asp and Tyr. Dietary crude protein linearly decreased plasma concentrations of Met, Thr, Glu+Gln and Gly. Sorghum inclusions linearly decreased plasma

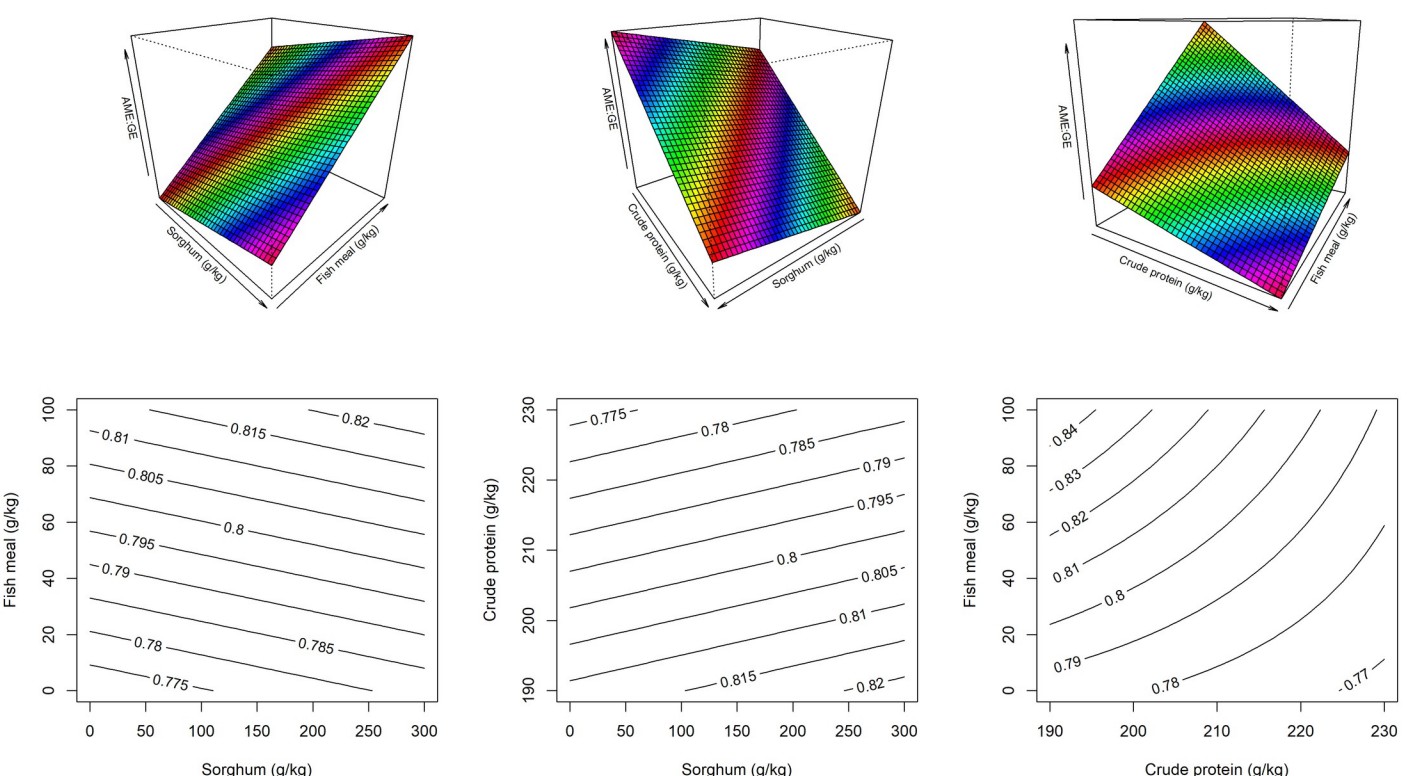

**Fig 5. The response surface designs and contour plots describing the impact of dietary crude protein level with sorghum and fishmeal inclusions on AME:GE ratio in broiler chickens in 27–29 days post-hatch (left–slice at 210 g/kg of CP; middle–slice at 5 g/kg of FM; right–slice at 150 g/kg of sorghum).**

concentrations of Leu, Val and Tyr. Fishmeal inclusions linearly increased plasma concentrations of His, Ile, Leu, Met, Phe, Thr, Val, Ala, Cys, Glu+Gln, Gly, Pro and Ser; but decreased plasma concentrations of Arg, Lys and Asn + Asp.

## Discussion

In the present study, dietary CP reductions from 230 to 210 and 190 g/kg linearly increased weight gain (r = 0.414, P < 0.01) and feed intake (r = 0.486, P < 0.001) without influencing FCR. Accordingly, birds offered 190 g/kg CP diets had 9.43% (2008 versus 1835 g/bird, P < 0.05) higher weight gain and 8.20% (3206 versus 2963 g/bird, P < 0.05) higher feed intake compared to birds offered 230 g/kg CP diets. However, the 190 g/kg CP diet represents a modest reduction in dietary CP for birds from 14 to 35 days post-hatch. Chrystal et al. [20] reported modest reduction in dietary CP increased weight gain by 8.22% (2396 versus 2214 g/bird) and reduced FCR 2.62% (1.453 versus 1.415) from 7–35 days post-hatch. Thus, birds have the potential to perform satisfactorily when offered moderately reduced-CP diets which was consistent with the present study. However, more tangible reductions in dietary CP usually compromise growth performance with an increase in carcass fat deposition [21, 22]. In the present study, on average, transition of dietary CP from 230 to 190 g/kg reduced soybean meal inclusions (56 versus 177 g/kg) whilst increasing NBAA inclusions (7.49 versus 19.8 g/kg). This outcome supports the rationale in Selle et al. [23] that replacing soybean meal with NBAA in

**Table 6. Effect of dietary treatments on apparent starch digestibility coefficients and disappearance rates in distal jejunum and distal ileum at 35 days post-hatch.**

| Diet | Digestibility coefficients | | Disappearance rates (g/bird/day) | |
|---|---|---|---|---|
| | Distal jejunum | Distal ileum | Distal jejunum | Distal ileum |
| 1A | 0.969 | 0.999 | 60.0 | 62.2 |
| 2B | 0.970 | 0.997 | 58.4 | 60.5 |
| 3C | 0.966 | 0.999 | 59.4 | 61.4 |
| 4D | 0.968 | 0.996 | 56.8 | 59.5 |
| 5E | 0.964 | 0.996 | 56.2 | 58.1 |
| 6F | 0.968 | 0.998 | 64.2 | 67.4 |
| 7G | 0.968 | 0.970 | 48.9 | 50.0 |
| 8H | 0.951 | 0.997 | 61.1 | 64.0 |
| 9I | 0.970 | 0.997 | 56.0 | 57.5 |
| 10J | 0.971 | 0.998 | 69.9 | 71.9 |
| 11K | 0.964 | 0.997 | 48.6 | 49.2 |
| 12L | 0.965 | 0.999 | 73.6 | 76.1 |
| 13M | 0.964 | 0.997 | 63.0 | 65.2 |
| **Crude protein (g/kg)** | | | | |
| 190 | 0.964 | 0.998 | 67.2 | 70.0 |
| 210 | 0.967 | 0.998 | 59.6 | 61.8 |
| 230 | 0.967 | 0.995 | 52.4 | 59.9 |
| Linear relationship (r =) | 0.094 | -0.284 | -0.800 | -0.812 |
| P-value | 0.455 | 0.027 | <0.0001 | <0.0001 |
| **Sorghum (g/kg)** | | | | |
| 0 | 0.966 | 0.998 | 59.6 | 62.4 |
| 150 | 0.963 | 0.995 | 58.7 | 60.1 |
| 300 | 0.970 | 0.998 | 61.2 | 63.1 |
| Linear relationship (r =) | 0.129 | 0.004 | 0.086 | 0.071 |
| P-value | 0.306 | 0.978 | 0.496 | 0.589 |
| **Fishmeal (g/kg)** | | | | |
| 0 | 0.964 | 0.995 | 56.4 | 58.4 |
| 50 | 0.967 | 0.998 | 62.2 | 64.6 |
| 100 | 0.967 | 0.998 | 59.9 | 62.0 |
| Linear relationship (r =) | 0.071 | 0.334 | 0.190 | 0.190 |
| P-value | 0.577 | 0.009 | 0.130 | 0.143 |

reduced-CP diets could be a promising strategy to reduce the chicken-meat industry's dependence on soybean meal. However, further reductions of dietary CP requires higher inclusions of NBAA as evident in Chrystal et al. [2]. This may generate amino acid imbalances at sites of protein synthesis; if so, increased deamination would generate ammonia and may compromise broiler growth performance [24].

Fishmeal is usually recognised as a prolific source of digestible amino acids in broiler chickens [25] and pigs [26] and was included in the present experiment to diversify the rate of protein digestion in order to investigate the influence of protein and starch digestive dynamics in diets containing different CP levels. However, it was not anticipated that the 100 g/kg fish meal inclusion would significantly compromise growth performance in the present study. This is in complete contrast to the positive growth performance responses generated by 175 g/kg fishmeal inclusions in sorghum-based broiler diets observed by Sydenham et al. [10]. Nevertheless, the unexpected impact of fishmeal is not without precedent. It was reported decades ago [27] that overheating fishmeal during the rendering process depressed the availability of amino

**Table 7. Effect of dietary treatments on apparent crude protein digestibility coefficients, disappearance rates and starch: Protein disappearance rate ratios in distal jejunum and distal ileum at 35 days post-hatch.**

| Diet | Digestibility coefficients | | Disappearance rates (g/bird/day) | | Starch: protein disappearance rate ratio | |
|---|---|---|---|---|---|---|
| | Distal jejunum | Distal ileum | Distal jejunum | Distal ileum | Distal jejunum | Distal ileum |
| 1A | 0.618 | 0.725 | 18.69 | 22.01 | 3.27 | 2.80 |
| 2B | 0.598 | 0.723 | 20.84 | 25.14 | 2.93 | 2.41 |
| 3C | 0.623 | 0.736 | 18.39 | 21.76 | 3.27 | 2.82 |
| 4D | 0.610 | 0.706 | 21.15 | 24.53 | 2.73 | 2.30 |
| 5E | 0.586 | 0.738 | 20.23 | 25.51 | 2.79 | 2.28 |
| 6F | 0.528 | 0.727 | 16.31 | 22.30 | 4.15 | 2.96 |
| 7G | 0.578 | 0.671 | 21.87 | 25.37 | 2.31 | 1.99 |
| 8H | 0.619 | 0.749 | 20.21 | 24.50 | 3.05 | 2.62 |
| 9I | 0.607 | 0.697 | 21.05 | 24.20 | 2.69 | 2.39 |
| 10J | 0.583 | 0.703 | 18.14 | 21.83 | 3.90 | 3.30 |
| 11K | 0.649 | 0.774 | 22.60 | 26.96 | 2.16 | 1.89. |
| 12L | 0.660 | 0.766 | 21.57 | 25.02 | 3.43 | 3.05 |
| 13M | 0.618 | 0.752 | 19.93 | 24.12 | 3.20 | 2.70 |
| **Crude protein (g/kg)** | | | | | | |
| 190 | 0.598 | 0.736 | 19.06 | 23.41 | 3.63 | 2.98 |
| 210 | 0.613 | 0.729 | 19.80 | 23.53 | 3.08 | 2.61 |
| 230 | 0.604 | 0.720 | 21.44 | 25.51 | 2.49 | 2.15 |
| Linear relationship (r =) | 0.033 | -0.128 | 0.292 | 0.354 | -0.639 | -0.779 |
| P-value | 0.796 | 0.321 | 0.019 | 0.004 | <0.0001 | <0.0001 |
| **Sorghum (g/kg)** | | | | | | |
| 0 | 0.636 | 0.746 | 20.93 | 24.57 | 2.90 | 2.56 |
| 150 | 0.586 | 0.728 | 19.71 | 24.38 | 3.10 | 2.49 |
| 300 | 0.602 | 0.712 | 19.68 | 23.29 | 3.20 | 2.72 |
| Linear relationship (r =) | -0.160 | -0.260 | -0.153 | -0.214 | 0.167 | 0.169 |
| P-value | 0.203 | 0.036 | 0.223 | 0.087 | 0.184 | 0.226 |
| **Fishmeal (g/kg)** | | | | | | |
| 0 | 0.601 | 0.713 | 21.01 | 24.89 | 2.76 | 2.33 |
| 50 | 0.623 | 0.738 | 20.66 | 24.44 | 3.08 | 2.70 |
| 100 | 0.589 | 0.732 | 18.41 | 22.89 | 3.37 | 2.70 |
| Linear relationship (r =) | -0.058 | 0.147 | -0.321 | -0.336 | 0.342 | 0.375 |
| P-value | 0.643 | 0.241 | 0.009 | 0.006 | 0.005 | 0.005 |

acids in broiler chickens. Lysine and aspartic acid, in particular, appear to be vulnerable to improper processing in both fishmeal and other protein meals [28–30]. In the present study, fishmeal inclusion negatively influenced plasma concentrations of Lys (r = -0.397, P = 0.001) and Asn + Asp (r = -0.478, P < 0.001). Unfortunately, due to the lack of variable impact of sorghum and fishmeal inclusions on distal jejunal starch and protein digestibilities, respectively, the relevant importance of starch and protein digestive dynamics in diets containing different CP was not tested as originally planned. However, attention needs to be drawn on inconsistent growth performance in broiler chickens when offered diets containing sorghum and fishmeal.

Sorghum has been associated with sub-optimal growth performance in broiler chickens, consequently, its inclusion in practical diets is often limited. However, in the present study, the substitution of wheat with sorghum did not influence broiler performance, which implies the two feed grains used in the present study were nutritionally equivalent where broilers offered diets with 0 and 300 g/kg sorghum generated comparable starch digestibility coefficients in

**Table 8. Effects of dietary treatments on free essential amino acid systemic plasma concentrations (µg/mL) at 34 days post-hatch.**

| Diet | Arg | His | Ile | Leu | Lys | Met | Phe | Thr | Trp | Val |
|------|-----|-----|-----|-----|-----|-----|-----|-----|-----|-----|
| 1A | 77.5 | 20.5 | 11.4 | 29.0 | 14.5 | 15.5 | 22.4 | 109.3 | 4.6 | 23.0 |
| 2B | 119.2 | 8.1 | 9.8 | 22.5 | 17.8 | 10.8 | 19.0 | 58.6 | 4.9 | 18.4 |
| 3C | 66.9 | 21.8 | 14.2 | 23.7 | 13.8 | 14.5 | 22.7 | 106.1 | 5.0 | 28.9 |
| 4D | 110.6 | 8.7 | 11.2 | 18.2 | 22.0 | 10.0 | 17.0 | 64.3 | 4.8 | 21.2 |
| 5E | 106.5 | 26.2 | 16.1 | 31.7 | 15.5 | 12.6 | 24.0 | 92.3 | 5.4 | 31.9 |
| 6F | 58.8 | 5.5 | 13.9 | 20.3 | 15.2 | 17.6 | 17.0 | 114.2 | 5.4 | 30.8 |
| 7G | 106.5 | 13.8 | 12.4 | 23.1 | 17.2 | 8.3 | 20.0 | 54.5 | 5.3 | 22.7 |
| 8H | 100.6 | 5.0 | 12.1 | 15.8 | 21.9 | 14.2 | 16.2 | 72.5 | 5.1 | 24.6 |
| 9I | 116.2 | 21.4 | 14.8 | 31.9 | 16.7 | 12.0 | 23.5 | 73.2 | 5.6 | 29.1 |
| 10J | 87.8 | 9.6 | 12.1 | 22.3 | 17.4 | 17.3 | 19.8 | 100.6 | 4.9 | 25.8 |
| 11K | 120.5 | 22.1 | 15.5 | 26.2 | 17.5 | 10.8 | 22.1 | 74.3 | 5.5 | 30.2 |
| 12L | 76.9 | 5.2 | 13.9 | 17.9 | 15.6 | 15.1 | 18.0 | 102.7 | 5.3 | 31.1 |
| 13M | 100.6 | 13.9 | 11.1 | 23.1 | 14.2 | 12.0 | 20.4 | 80.5 | 5.00 | 21.9 |
| **Linear relationships** | | | | | | | | | | |
| **Crude protein (g/kg)** | | | | | | | | | | |
| 190 | 81.0 | 6.3 | 13.0 | 19.1 | 17.5 | 16.1 | 17.8 | 97.5 | 5.2 | 28.1 |
| 210 | 95.0 | 14.6 | 11.5 | 23.3 | 16.5 | 12.5 | 20.3[b] | 83.8 | 4.9 | 22.6 |
| 230 | 112.4 | 20.9 | 14.7 | 28.2 | 16.7 | 10.9 | 22.4 | 73.6 | 5.4 | 28.5 |
| r = | 0.453 | 0.726 | 0.267 | 0.661 | -0.066 | -0.598 | 0.614 | -0.424 | 0.177 | 0.029 |
| P = | <0.0001 | <0.001 | 0.032 | <0.001 | 0.602 | <0.0001 | <0.001 | <0.001 | 0.160 | 0.821 |
| **Sorghum (g/kg)** | | | | | | | | | | |
| 0 | 93.7 | 14.5 | 13.7 | 21.5 | 17.2 | 12.6 | 20.0 | 86.8 | 5.1 | 27.8 |
| 150 | 94.6 | 12.9 | 13.1 | 22.8 | 16.8 | 13.0 | 19.5 | 82.8 | 5.2 | 26.4 |
| 300 | 100.2 | 14.9 | 12.0 | 26.4 | 16.6 | 13.9 | 21.2 | 85.4 | 5.0 | 24.1 |
| r = | 0.093 | 0.022 | -0.258 | 0.357 | -0.047 | 0.154 | 0.159 | -0.025 | -0.081 | -0.275 |
| P = | 0.460 | 0.864 | 0.038 | 0.004 | 0.711 | 0.221 | 0.206 | 0.842 | 0.521 | 0.026 |
| **Fishmeal (g/kg)** | | | | | | | | | | |
| 0 | 109.2 | 8.9 | 11.4 | 19.9 | 19.7 | 10.8 | 18.1 | 62.5 | 5.0 | 21.7 |
| 50 | 100.4 | 14.5 | 13.5 | 24.3 | 16.3 | 13.4 | 20.7 | 86.3 | 5.3 | 27.6 |
| 100 | 77.4 | 18.5 | 13.9 | 26.2 | 14.7 | 15.1 | 21.5 | 105.5 | 5.1 | 28.6 |
| r = | -0.459 | 0.477 | 0.393 | 0.450 | -0.397 | 0.492 | 0.460 | 0.761 | 0.055 | 0.504 |
| P = | <0.001 | <0.001 | 0.001 | 0.0002 | 0.001 | <0.001 | 0.001 | <0.001 | 0.665 | <0.001 |

distal jejunum (0.966 versus 0.970). This suggests that the starch digestibility/energy utilization of sorghum used in the present study was of an unusually high order. Wheat is commonly recognised as a better feed grain under Australian conditions, but this is not necessarily always the case. The performance of broilers offered maize-, sorghum- and wheat-based diets, without and with exogenous phytase, was compared from 1 to 27 days post-hatch in Liu et al. [31]. As main effects, sorghum supported significantly better weight gain by 6.70% (1338 versus 1254 g/bird) and FCR by 3.60% (1.471 versus 1.526) than wheat-based diets. Alternatively, Moss et al. [32] compared two sorghum-based diets with two wheat-based diets offered to broilers from 1 to 35 days post-hatch. While average weight gains were nearly identical (2670 versus 2676 g/bird), birds offered wheat-based diets enjoyed an advantage of 3.41% (1.415 versus 1.465) in FCR. Thus, the relative nutritional values of the two feed grains are sufficiently variable that the outcome from any one comparison cannot be predicted with any accuracy. The comparable quality of sorghum starch to wheat starch in the present study failed to generate different starch digestion rates.

**Table 9. Effects of dietary treatments on free non-essential and total amino acid systemic plasma concentrations (µg/mL) at 34 days post-hatch.**

| Diet | Ala | Asn+Asp | Cys | Glu+Gln | Gly | Pro | Ser | Tyr | Total |
|---|---|---|---|---|---|---|---|---|---|
| 1A | 78.9 | 94.7 | 16.0 | 242.2 | 138.9 | 72.4 | 83.2 | 40.7 | 1036.0 |
| 2B | 65.2 | 137.1 | 15.9 | 190.8 | 47.7 | 41.7 | 54.7 | 41.7 | 775.2 |
| 3C | 72.8 | 79.9 | 18.5 | 277.0 | 127.3 | 81.8 | 76.5 | 37.1 | 1042.2 |
| 4D | 59.8 | 130.4 | 15.1 | 196.4 | 43.8 | 45.6 | 54.4 | 34.2 | 763.4 |
| 5E | 77.7 | 122.4 | 17.4 | 223.1 | 105.4 | 78.7 | 80.4 | 43.0 | 1025.2 |
| 6F | 78.7 | 73.9 | 16.7 | 274.8 | 165.4 | 57.7 | 89.6 | 29.7 | 1044.9 |
| 7G | 58.7 | 123.1 | 14.8 | 181.0 | 46.9 | 45.0 | 54.3 | 38.3 | 751.8 |
| 8H | 68.5 | 114.5 | 14.7 | 221.4 | 48.9 | 48.3 | 49.6 | 28.6 | 793.0 |
| 9I | 77.8 | 134.7 | 15.8 | 202.8 | 76.0 | 60.1 | 58.1 | 47.3 | 919.3 |
| 10J | 74.5 | 105.5 | 15.6 | 243.7 | 110.2 | 57.8 | 72.5 | 30.9 | 954.8 |
| 11K | 71.2 | 137.4 | 16.9 | 202.3 | 71.4 | 66.3 | 64.8 | 39.7 | 912.2 |
| 12L | 72.6 | 97.3 | 17.0 | 283.5 | 100.5 | 59.4 | 72.3 | 26.8 | 963.8 |
| 13M | 68.3 | 116.5 | 15.7 | 208.2 | 81.3 | 59.2 | 61.1 | 37.5 | 864.6 |
| **Linear relationships** | | | | | | | | | |
| **Crude protein (g/kg)** | | | | | | | | | |
| 190 | 73.6 | 97.8 | 16.0 | 255.9 | 106.3 | 55.8 | 71.0 | 29.0 | 939.1 |
| 210 | 69.0 | 111.7 | 16.4 | 222.9 | 87.8 | 60.1 | 66.0 | 38.4 | 896.4 |
| 230 | 71.4 | 129.4 | 16.2 | 202.3 | 74.9 | 62.5 | 64.4 | 42.1 | 902.1 |
| r = | -0.076 | 0.448 | 0.054 | -0.434 | -0.301 | 0.175 | -0.187 | 0.736 | -0.105 |
| P = | 0.135 | 0.0002 | 0.671 | 0.0003 | 0.015 | 0.164 | 0.135 | <0.001 | 0.405 |
| **Sorghum (g/kg)** | | | | | | | | | |
| 0 | 69.1 | 111.3 | 16.8 | 239.8 | 85.7 | 62.3 | 67.0 | 34.4 | 920.5 |
| 150 | 70.4 | 110.1 | 15.9 | 221.7 | 89.6 | 57.8 | 67.0 | 35.4 | 895.9 |
| 300 | 74.1 | 118.0 | 16.0 | 219.9 | 93.2 | 58.0 | 67.1 | 40.2 | 921.3 |
| r = | 0.171 | 0.095 | -0.197 | -0.162 | -0.197 | -0.137 | 0.003 | 0.323 | 0.002 |
| P = | 0.173 | 0.450 | 0.116 | 0.198 | 0.572 | 0.276 | 0.982 | 0.009 | 0.985 |
| **Fishmeal (g/kg)** | | | | | | | | | |
| 0 | 63.0 | 126.3 | 15.1 | 197.4 | 46.8 | 45.1 | 53.3 | 35.7 | 770.8 |
| 50 | 72.9 | 118.3 | 16.2 | 228.1 | 87.9 | 60.6 | 65.8 | 36.5 | 922.9 |
| 100 | 77.0 | 92.7 | 17.4 | 254.3 | 134.3 | 72.7 | 82.4 | 37.6 | 1037.2 |
| r = | 0.479 | -0.478 | 0.554 | 0.461 | 0.839 | 0.715 | 0.830 | 0.110 | 0.756 |
| P = | <0.001 | <0.0001 | <0.001 | 0.0001 | <0.001 | <0.000 | <0.001 | 0.384 | <0.001 |

The Box-Behnken design is challenged in the present study because there are additional variations in dietary compositions to the three factors being evaluated. Concentrations of soybean meal ranged from 20.1 to 253 g/kg and NBAA from 6.13 to 19.90 g/kg across the 13 dietary treatments (Table 2). Interestingly, significant multiple linear regressions were detected for feed intakes (r = 0.746; P < 0.0001) and weight gains (r = 0.860; P < 0.0001) when soybean meal and NBAA were considered in addition to fishmeal. The equations are as follows:

$$\text{Feed intake}_{\text{(g/bird)}} = 2402 - 0.247 \times \text{FM}_{\text{(g/kg)}} + 1.718 \times \text{SBM}_{\text{(g/kg)}} + 36.36 \times \text{NBAA}_{\text{(g/kg)}}$$

$$\text{Weight gain}_{\text{(g/bird)}} = 1849 - 3.016 \times \text{FM}_{\text{(g/kg)}} + 0.05 \times \text{SBM}_{\text{(g/kg)}} + 14.5 \times \text{NBAA}_{\text{(g/kg)}}.$$

Thus, in both instances, inclusions of soybean meal and NBAA promoted feed intakes and weight gains; whereas, fishmeal inclusions had negative impacts on growth performance.

## Conclusions

Growth performance were compromised by higher fishmeal inclusion and was not influenced by sorghum substitution. Both fishmeal and sorghum inclusions did not alter protein and starch digestion rate in broiler chickens; however, it is evident that moderate reductions in dietary CP could advantage broiler growth performance.

## Supporting information

**S1 Raw data.**
(XLSX)

## Acknowledgments

We would like to thank Ms Joy Gill, Ms Kylie Warr, Mr Duwei Chen and Mr Peter Bird of the Poultry Research Foundation within the University of Sydney for their invaluable technical assistance.

## Author Contributions

**Conceptualization:** Peter V. Chrystal, Peter H. Selle, Sonia Y. Liu.

**Data curation:** Shemil P. Macelline, Shiva Greenhalgh, Mehdi Toghyani.

**Formal analysis:** Shemil P. Macelline, Peter H. Selle.

**Funding acquisition:** Sonia Y. Liu.

**Investigation:** Shemil P. Macelline, Shiva Greenhalgh, Mehdi Toghyani, Peter H. Selle.

**Methodology:** Shemil P. Macelline, Peter V. Chrystal, Peter H. Selle.

**Project administration:** Sonia Y. Liu.

**Supervision:** Peter H. Selle, Sonia Y. Liu.

**Visualization:** Shemil P. Macelline.

**Writing – original draft:** Shemil P. Macelline, Peter H. Selle, Sonia Y. Liu.

**Writing – review & editing:** Shemil P. Macelline, Peter V. Chrystal, Shiva Greenhalgh, Mehdi Toghyani, Peter H. Selle, Sonia Y. Liu.

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
