## [Decision Letter · Decision Letter 0]

6 Oct 2021

PONE-D-21-26429Box-Behnken evaluation of dietary crude protein concentrations, fishmeal, and sorghum inclusions in wheat-based diets in broiler chickens from 14 to 35 days post-hatchPLOS ONE

Dear Dr. Liu,

Thank you for submitting your manuscript to PLOS ONE. After careful consideration, we feel that it has merit but does not fully meet PLOS ONE’s publication criteria as it currently stands. Therefore, we invite you to submit a revised version of the manuscript that addresses the points raised during the review process.

The content is not clear and is difficult for the reader to understand main points in the sections such as the introduction and the flexible methodologies. The authors should do a better job to improve in materials and methods. The quality of all figures needs to improve and to describe in detail.

We look forward to receiving your revised manuscript.

Kind regards,

Arda Yildirim, Ph.D.

Academic Editor

PLOS ONE

Journal Requirements:

“We would like to acknowledge the financial support provided by Australian Government Research and Training Program (RTP) Scholarship for the PhD candidature of Mr Shemil Macelline.”

We note that you have provided additional information within the Acknowledgements. Please note that funding information should not appear in the Acknowledgments section or other areas of your manuscript. We will only publish funding information present in the Funding Statement section of the online submission form.

“This study is funded by Australian Research Council Discovery Early Career Researcher Award (DE190101364) and the award supports Dr Sonia Liu’s salary and the feeding study. The funder did not have any additional role in the study design, data collection and analysis, decision to publish or preparation of the manuscript.”

Additional Editor Comments:

This manuscript deals with an interesting and important topic in poultry nutrition. Nevertheless there are still some points of concern from the reviewers, before the manuscript can be accepted for publication. Please focus on animal material and method, Tables and figures. As there are some points unclear to me regarding the trial execution, sampling and statements in the manuscript, I recommend major revision.

Reviewers' comments:

Reviewer's Responses to Questions

**Comments to the Author**

1. Is the manuscript technically sound, and do the data support the conclusions?

Reviewer #1: Yes

Reviewer #2: Yes

Reviewer #3: Yes

2. Has the statistical analysis been performed appropriately and rigorously? 

Reviewer #1: Yes

Reviewer #2: Yes

Reviewer #3: Yes

3. Have the authors made all data underlying the findings in their manuscript fully available?

Reviewer #1: Yes

Reviewer #2: Yes

Reviewer #3: Yes

4. Is the manuscript presented in an intelligible fashion and written in standard English?

Reviewer #1: Yes

Reviewer #2: Yes

Reviewer #3: Yes

5. Review Comments to the Author

Reviewer #1: The paper is interesting. I am not familiar to Box-Behnke design and I think you should write more of this design. How were the number of the treatments decided? I was also wodering was the amino acid balance similar in all the diets. Do you have any information of the processing conditions of fish meal. It is surprised be that fish meal lowered the weight gain.

Reviewer #2: The manuscript titled ‘Box-Behnken evaluation of dietary crude protein concentrations, fishmeal, and sorghum inclusions in wheat-based diets in broiler chickens from 14 to 35 days post-hatch’ has been designed well to elucidate the effect of providing high and low levels of crude protein, sorghum, and fish meal compared with frequently used levels in the diet on the performance and nutrient utilization in growing-finishing broilers. Overall, the authors have done nice work, but there are a few important pieces of feedback for the improvement of the conclusiveness of this study. The addressing of these comments should be considered before the manuscript would be deemed acceptable for publication.

1. It would be better to present digestible amino acids value for at least major limiting amino acids as well in the nutrient composition. Since crystalline (non-bound) amino acids are used to balance the essential amino acids in the diet, it would change the level of digestible amino acids in the formulated feed. The declaration of the digestible amino acids in the diets used for this study is vital because it is a complex design and shows the predicted optimum composition of feed based on the best-fit regression model. This study is not only evaluating different ingredients but is also adding crude protein in the model and due to the variation in the digestibility of amino acids of the ingredients used in this study, it could make a certain conclusion about the effect of crude protein in the diet on growth performance which may not necessarily be true. Some previous studies have already shown that broilers grown on a low crude protein diet can have better performance. However, the information about the digestible value will prove that even high protein in the diet after a certain inclusion level may not improve the performance of broilers.

2. The second question is about the digestible amino acid value of the fish meal. The authors could provide the digestible crude protein and amino acids value of the fish meal used in this study as the digestible profile of this product can be substantially variable depending on source and processing methods.

3. Should make it Box-Behnken design or experimental model in the title. The phrase 'Box-Behnken evaluation' doesn’t sound appropriate.

4. Line 41-43: Negative opinion for soybean production should not be put forward without it being the theme of discussion. Securing sources of protein could also lead to overfishing, excess petroleum extraction etc.

5. Line 47: Should be footpad lesion scores.

6. Table 2 and rest of tables: better mention choline chloride. A common name would be better in the ingredient list. OR define in the caption below the table. Also, provide more information in the description caption of tables such as what the treatments from 1A to 13 M mean. The tables and figures should be standalone.

7. Line 123: Change ‘-6-hour-off’ to ‘-6-hours-off’.

8. Line 196: Change ‘weigh’ to ‘weight’ in the equation.

9. Line 305: Change ‘compromised’ to ‘compromise’.

Reviewer #3: Dear Authors,

I have a series of questions that I consider necessary to address in the present manuscript (Discussion) and that may be useful to you for future studies as well.

Main comments:

- Fishmeal: the justification to study “fishmeal” as main factor seems not totally well justified. It does not appear to be a more sustainable protein source compared to SBM and so helping to have a more sustainable meat chicken meat production as stated in L41-43.

o Moreover, with 100 g/kg fishmeal, canola meal was also partially replaced, so that would apparently induce an even higher environmental impact.

o Was fishmeal quality measured, and compared to previous studies (L301-311).

- The interaction between starch and protein digestive dynamics has been previously studied by your group. The Box-Behnken design studying these three factors (CP level, fish meal level and sorghum level) is not well justified, and hypothesis for their potential interaction are not well described in the introduction and discussion sections.

- Conclusion. Form L341 and 350 it is not a conclusion but a results summary. The only conclusion written is in L351-352.

Study design:

- Why was the study conducted from day 14 onwards and not from day 0 until the end of growing period? Why is your design more relevant than having different feeding phases.

- Why was phytase not used in the experimental diets!?

- Were test feedstuffs analysed before diet reformulation and production? I would expect that at least wheat, sorghum, SBM and fishmeal were analysed. These results should be presented here as well.

- Why only 5 replicates were used per treatment? This number seems very small.

- Why was excreta collected from 27-29 days and intestinal contents on day 34? The age difference could have make the comparison of these parameters not possible.

Additional comments:

- L144: 20 amino acids. Why Asn+Asp and Glu+Gln are presented together as a sum?

- L149-150: reference method?

- L151: what vein?

- L151: Why were (390!) birds (individually) sacrificed by intravenous injection instead of by using CO2 per pen? The individual sacrifice should have cost a large amount of time and seems not to be that convenient.

- L152: Why was ileal content collected from distal jejunum + distal ileum to determine ileal digestibility? This is a rather large section at this bird age.

- L154-158: repetititve.

- L157: it seems that the contents of both distal jejunum and distal ileuam were pooled.

- L161: it is okay to repeat reference method.

- L163: What 4 sites??? In Table 6 and 7 there are only 2 sites presented.

- L168: average Feed intake 14-35d?

- L203-206: no interaction?

- L207: R2: exactly the same as for BWG?

- L214: P-value?

- L222: interactions?

- L238: Why is ME:GE ratio a relevant parameter?

- L260-263: Why not explanation of main factors?

- L270-274: Not presented? What is the relevance of these results?

Table 2:

- Maize starch was used as an ingredient. So, why not targeting the same starch content in all diets? Sand was used as a filler (only in diet 6F?).

- Glycine: was it L-glycine?

- C5H12ClNO => Choline chloride

- I would suggest to merge tables 2 and 3. If that does not happen, I would suggest to present “Total NBAA”, “Starch specs” and “Starch analysed” in Table 3; “Starch analysed” is currently presented in both Table 2 and 3.

Table 3:

- Present (calculated and analysed) C Fat and CFibre levels.

- It would be more interesting to present digestible AA contents rather than total.

- dEB: why was it not corrected by using K-carbonate or another K source? Discuss the effect of dEB on your study.

Table 5:

- There is a redundant row Under “Sorghum” main effect.

Table 7: P-values of CP factor?

Figures: add units of the response parameters. e.g. %, g, use the same units as on the Tables. Image quality is poor.

Figure 2 is the same as in Figure 1 for BWG!!!! The figure of FI is not shown!!!

6. PLOS authors have the option to publish the peer review history of their article (what does this mean?). If published, this will include your full peer review and any attached files.

Reviewer #1: No

Reviewer #2: No

Reviewer #3: No

---

## [Author Response · Author response to Decision Letter 0]

24 Oct 2021

Journal Requirements:

Updated 

“We would like to acknowledge the financial support provided by Australian Government Research and Training Program (RTP) Scholarship for the PhD candidature of Mr Shemil Macelline.”

We note that you have provided additional information within the Acknowledgements. Please note that funding information should not appear in the Acknowledgments section or other areas of your manuscript. We will only publish funding information present in the Funding Statement section of the online submission form.

“This study is funded by Australian Research Council Discovery Early Career Researcher Award (DE190101364) and the award supports Dr Sonia Liu’s salary and the feeding study. The funder did not have any additional role in the study design, data collection and analysis, decision to publish or preparation of the manuscript.”

Updated in the cover letter

 Updated

Additional Editor Comments:

This manuscript deals with an interesting and important topic in poultry nutrition. Nevertheless there are still some points of concern from the reviewers, before the manuscript can be accepted for publication. Please focus on animal material and method, Tables and figures. As there are some points unclear to me regarding the trial execution, sampling and statements in the manuscript, I recommend major revision.

Thank you, we have address all reviewers’ comments.

Review Comments to the Author

Reviewer #1: The paper is interesting. I am not familiar to Box-Behnken design and I think you should write more of this design. How were the number of the treatments decided? I was also wondering was the amino acid balance similar in all the diets. Do you have any information of the processing conditions of fish meal. It is surprised be that fish meal lowered the weight gain.

Thank you, the below is included in the Introduction,

Box-Behnken design (BBD) is a multivariate optimization design with the advantage of testing multiple nutrients simultaneously with less number of treatments (De Leon et al., 2010). It was previously used to optimise digestive dynamics and compare relative importance of dietary factors (Liu et al., 2016; 2019). Therefore, the experimental diets in the present study contained three levels of dietary CP (190, 210, 230 g/kg), three levels of fishmeal (0, 50, 100 g/kg), and three levels of sorghum (0, 150, 300 g/kg) and response surface was plotted to visualise experimental results.

BBD is a classic way of optimisation. Once experimental factors are confirmed to be influential by factorial design, BBD can help to find the best numerical combinations of these variables. It is widely used in engineering. Unfortunately, we do not have processing and source information of fishmeal which we regret. It is certainly a surprise fishmeal depressed performance here.

Reviewer #2: The manuscript titled ‘Box-Behnken evaluation of dietary crude protein concentrations, fishmeal, and sorghum inclusions in wheat-based diets in broiler chickens from 14 to 35 days post-hatch’ has been designed well to elucidate the effect of providing high and low levels of crude protein, sorghum, and fish meal compared with frequently used levels in the diet on the performance and nutrient utilization in growing-finishing broilers. Overall, the authors have done nice work, but there are a few important pieces of feedback for the improvement of the conclusiveness of this study. The addressing of these comments should be considered before the manuscript would be deemed acceptable for publication.

1. It would be better to present digestible amino acids value for at least major limiting amino acids as well in the nutrient composition. Since crystalline (non-bound) amino acids are used to balance the essential amino acids in the diet, it would change the level of digestible amino acids in the formulated feed. The declaration of the digestible amino acids in the diets used for this study is vital because it is a complex design and shows the predicted optimum composition of feed based on the best-fit regression model. This study is not only evaluating different ingredients but is also adding crude protein in the model and due to the variation in the digestibility of amino acids of the ingredients used in this study, it could make a certain conclusion about the effect of crude protein in the diet on growth performance which may not necessarily be true. Some previous studies have already shown that broilers grown on a low crude protein diet can have better performance. However, the information about the digestible value will prove that even high protein in the diet after a certain inclusion level may not improve the performance of broilers.

Totally agreed, we apologise for not pointing this out in the original submission. All experimental diets were formulated to digestible AA basis, Table 3 has been updated to reflect this. The amino acids formulated to constant levels (digestible basis) include TSAA, Thr, Arg and Gly-equivalent. The amino acids formulated to minimal levels (digestible basis) include Trp, Ile and Val. 

2. The second question is about the digestible amino acid value of the fish meal. The authors could provide the digestible crude protein and amino acids value of the fish meal used in this study as the digestible profile of this product can be substantially variable depending on source and processing methods.

We regret the fishmeal we used in the present study was not analysed by NIR for dig AAs as fishmeal is not routinely used in broiler diets. It is surprising in the present study fishmeal did not perform as well as our previous study in 2016. The below was used for formulation based on our historical wet chemistry data in the 2016 experiment. 

 % % % % % %

Ingredients Nitrogen Protein Factor Starch Ca P Na

Fish meal 10.175 63.594 6.250 0.105 6.807 3.604 0.655

Liu, SY, Sydenham, CJ, Selle, PH (2016) Feed access to, and inclusions of fishmeal and corn starch in, sorghum-based broiler diets influence growth performance and nutrient utilisation as assessed by the Box-Behnken response surface design. Animal Feed Science and Technology 220, 46-56.

3. Should make it Box-Behnken design or experimental model in the title. The phrase 'Box-Behnken evaluation' doesn’t sound appropriate.

This title is updated to “Evaluation of dietary crude protein concentrations, fishmeal, and sorghum inclusions in broiler chickens offered wheat-based diet via Box-Behnken response surface design”

4. Line 41-43: Negative opinion for soybean production should not be put forward without it being the theme of discussion. Securing sources of protein could also lead to overfishing, excess petroleum extraction etc.

Deleted

5. Line 47: Should be footpad lesion scores.

Corrected

6. Table 2 and rest of tables: better mention choline chloride. A common name would be better in the ingredient list. OR define in the caption below the table. Also, provide more information in the description caption of tables such as what the treatments from 1A to 13 M mean. The tables and figures should be standalone.

Table 2 is updated; however, we are not sure whether it is practical to include footnote describing 13 treatments because this is different from factorial design. This is the reason table 1 was included.

 7. Line 123: Change ‘-6-hour-off’ to ‘-6-hours-off’.

Corrected 

8. Line 196: Change ‘weigh’ to ‘weight’ in the equation.

Corrected

9. Line 305: Change ‘compromised’ to ‘compromise’.

Corrected

Reviewer #3: Dear Authors,

I have a series of questions that I consider necessary to address in the present manuscript (Discussion) and that may be useful to you for future studies as well.

Main comments:

- Fishmeal: the justification to study “fishmeal” as main factor seems not totally well justified. It does not appear to be a more sustainable protein source compared to SBM and so helping to have a more sustainable meat chicken meat production as stated in L41-43. Moreover, with 100 g/kg fishmeal, canola meal was also partially replaced, so that would apparently induce an even higher environmental impact.

Totally agree L41-43 is deleted and the first paragraph of Introduction is included as below,

“Developing reduced crude protein (CP) diets where soybean meal particularly replaced with supplementary amino acids is a promising nutritional strategy to achieve sustainable chicken meat production with reduced nitrogen excretion and improved bird welfare. Dietary CP reduction from 220 to 160 g/kg reduced nitrogen excretion by 35% and dietary CP reduction from 198 to 169 g/kg reduced foot-pad lesion scores by 59% in broiler chickens as reviewed by Greenhalgh et al. [1]. Moreover, a dietary CP reduction from 222 to 165 g/kg reduced soybean meal inclusions by 74% in Chrystal et al. [3]. However, broiler chickens responded to reduced CP diets inconsistently; for instance, broilers offered maize-based, reduced-CP diets improved growth performance in comparison to standard CP diets; whereas, broilers offered wheat-based, reduced-CP diets displayed inferior growth performance [3].”

“Previous studies reported the relevance of protein digestion rate in broiler diet using casein and whey protein concentrate. It is intended to apply a more practical feed ingredient in the present study to test the relevance of starch and protein digestive dynamics.”

The intention of this study is not to use fishmeal to replace SBM, it is preliminary test the relevance of starch and protein digestive dynamics in reduced CP diets. We previously reported the relevance by using whey-protein, and we gradually moved to a semi-practical ingredient in this study. Unfortunately, fishmeal in the present study did not perform as a rapidly digestible protein source as we expected, which raised another interesting question of consistency of fishmeal quality. 

Was fishmeal quality measured, and compared to previous studies (L301-311).

Unfortunately and regrettably not because of the challenge of sourcing fishmeal on time. Fishmeal is not routinely used in broiler diets presently.

- The interaction between starch and protein digestive dynamics has been previously studied by your group. The Box-Behnken design studying these three factors (CP level, fish meal level and sorghum level) is not well justified, and hypothesis for their potential interaction are not well described in the introduction and discussion sections.

The below is included in Introduction:

“Previous studies reported the relevance of protein digestion rate in broiler diet using casein and whey protein concentrate. It is intended to apply a more practical feed ingredient in the present study to test the relevance of starch and protein digestive dynamics. The hypothesis was both starch and protein digestive dynamics, which was reflected by variable dietary fishmeal and sorghum inclusions, would influence growth performance and nutrient utilisation; moreover, it was expected the impact of starch and protein digestive dynamics was more pronounced in diets containing lower CP and higher NBAA.” 

- Conclusion. Form L341 and 350 it is not a conclusion but a results summary. The only conclusion written is in L351-352.

Conclusion is updated as below

“Growth performance were compromised by higher fishmeal inclusion and was not influenced by sorghum substitution. Both fishmeal and sorghum inclusions did not alter protein and starch digestion rate in broiler chickens; however, it is evident that moderate reductions in dietary CP could advantage broiler growth performance.” 

Study design:

- Why was the study conducted from day 14 onwards and not from day 0 until the end of growing period? Why is your design more relevant than having different feeding phases.

It is for the below reasons:

(1) Performance in younger chicks (starter phase) could be confounded by breeder flock and hatchery conditions.

(2) Significant number of additional treatments would be required to truly test phase effect unless only cumulative response was determined. 

(3) This one is a practical/commercial concern. For reducing dietary crude protein, it is more profitable and less risk in grower and finisher phases, because they have much higher feed consumption and more resilient digestive and immune system. 

- Why was phytase not used in the experimental diets!?

It is not used because fishmeal was included at variable levels. Fishmeal contains high levels of mineral, the below is a historical (2016) wet chemistry analysis, we do not want to confound the study by using phytase with mineral matrix and worrying about inconsistent activity recovery and variation of dietary phytate levels. 

 % % % % % %

Ingredients Nitrogen Protein Factor Starch Ca P Na

Fish meal 10.175 63.594 6.250 0.105 6.807 3.604 0.655

- Were test feedstuffs analysed before diet reformulation and production? I would expect that at least wheat, sorghum, SBM and fishmeal were analysed. These results should be presented here as well.

Wheat, sorghum and SBM were tested before formulation and unfortunately, fishmeal was not tested due to the challenge of sourcing it. The results were not included because dig AAs used for formulation is analysed by NIR which is industry standard but we were suggested previously by other reviewers not to report it. 

However, the below is included in M&M,

“The nutritionally equivalent diets were formulated based on near-infrared spectroscopy (NIR) of wheat, sorghum and soybean meal using the AMINONir® Advanced program (Evonik Nutrition & Care GmbH, Hanau, Germany).”

- Why only 5 replicates were used per treatment? This number seems very small.

This is typical for BBD which emphasises the number of treatments (dots on the surface) is more critical than number of replications for each dot.

- Why was excreta collected from 27-29 days and intestinal contents on day 34? The age difference could have make the comparison of these parameters not possible.

We agree that age would influence AME and N retention results, and this is why we conduct collection between 27-29 days. In the literature, this is the age nutrient utilisation was reported and most matrix value used in formulation is quantified around 28 days post-hatch. The collection of intestinal content is for quantification of digestibility. 

Additional comments:

- L144: 20 amino acids. Why Asn+Asp and Glu+Gln are presented together as a sum?

This is to do with amino acid analysis method as they are not separated by ultra-performance liquid chromatography.

- L149-150: reference method?

Included

Cohen, SA (2001) Amino acid analysis using precolumn derivatisation with 6-aminoquinolyl-N-hydroxysuccinimidyl carbamate. Methods in Molecular Biology 159, 39-47.

- L151: what vein?

Updated to “brachial vein”

- L151: Why were (390!) birds (individually) sacrificed by intravenous injection instead of by using CO2 per pen? The individual sacrifice should have cost a large amount of time and seems not to be that convenient.

This is the routine method approved by our Ethic Committee

- L152: Why was ileal content collected from distal jejunum + distal ileum to determine ileal digestibility? This is a rather large section at this bird age.

Corrected to “digesta samples were collected in their entirety from the distal jejunum and distal ileum to determine apparent digestibility coefficients protein (N) and starch in the distal jejunum and ileum.”

- L154-158: repetititve.

Corrected

- L157: it seems that the contents of both distal jejunum and distal ileuam were pooled.

Yes, jejunal samples from the same cage were pooled and ileal samples from the same cage were pooled.

- L161: it is okay to repeat reference method.

Okay 

- L163: What 4 sites??? In Table 6 and 7 there are only 2 sites presented.

Corrected

- L168: average Feed intake 14-35d?

Updated to “…The overall average weight gain and feed intake for all treatments from 14 to 35 days post-hatch …”

- L203-206: no interaction?

No interaction here, non-significant term is excluded in the equation.

- L207: R2: exactly the same as for BWG?

Updated

- L214: P-value?

Included

- L222: interactions?

No interaction here, non-significant term is excluded in the equation

- L238: Why is ME:GE ratio a relevant parameter?

We have seen in the past better correlation between performance and ME:GE than with AME per se. This original idea is from 

Black, JL, Hughes, RJ, Nielsen, SG, Tredrea, AM, MacAlpine, R, Van Barneveld, RJ (2005) The energy value of cereal grains, particularly wheat and sorghum, for poultry. Proceedings of Australian Poultry Science Symposium 17, 21-29.

 - L260-263: Why not explanation of main factors?

Good suggestion, thank you and now they are included as shown in the track-change version

- L270-274: Not presented? What is the relevance of these results?

Deleted and updated as

“The impact of dietary treatments on free amino acids concentrations in the systemic plasma is shown in Table 8A and 8B. Dietary crude protein linearly increased plasma concentrations of Arg, His, Ile, Leu, Phe, Asn + Asp and Tyr. Dietary crude protein linearly decreased plasma concentrations of Met, Thr, Glu + Gln and Gly. Sorghum inclusions linearly decreased plasma concentrations of Leu, Val and Tyr. Fishmeal inclusions linearly increased plasma concentrations of His, Ile, Leu, Met, Phe, Thr, Val, Ala, Cys, Glu + Gln, Gly, Pro and Ser; but decreased plasma concentrations of Arg, Lys and Asn + Asp.”

Table 2:

- Maize starch was used as an ingredient. So, why not targeting the same starch content in all diets? Sand was used as a filler (only in diet 6F?).

The crude protein levels also changed, if starch content is fixed. Low CP diet will draw fat to formulate iso-energetic diets. Consequently, these diets will have much higher level of filler (> 10%) which is impossible to pellet.

- Glycine: was it L-glycine?

There is no L, D-form of Glycine

- C5H12ClNO => Choline chloride

Corrected

- I would suggest to merge tables 2 and 3. If that does not happen, I would suggest to present “Total NBAA”, “Starch specs” and “Starch analysed” in Table 3; “Starch analysed” is currently presented in both Table 2 and 3.

Only showing Starch and Total NBAA in Table 3.

Table 3:

- Present (calculated and analysed) C Fat and CFibre levels.

Calculated crude fat and fibre values are included in Table 3

- It would be more interesting to present digestible AA contents rather than total.

Corrected

- dEB: why was it not corrected by using K-carbonate or another K source? Discuss the effect of dEB on your study.

The impact of DEB on performance of reduced CP diets were reported in Chrystal et al. (2020) where dropping DEB from 230 to 156 mEq/kg in reduced CP diet did not influence growth performance, energy utilisation and protein digestibilities.

Chrystal, PV, Moss, AF, Khoddami, A, Naranjo, VD, Selle , PH, Liu, SY (2020) Effects of reduced crude protein levels, dietary electrolyte balance, and energy density on the performance of broiler chickens offered maize-based diets with evaluations of starch, protein, and amino acid metabolism. Poultry Science 99, 1421-1431.

Table 5:

- There is a redundant row Under “Sorghum” main effect.

Deleted

Table 7: P-values of CP factor?

Included

Figures: add units of the response parameters. e.g. %, g, use the same units as on the Tables. Image quality is poor.

All updated

Figure 2 is the same as in Figure 1 for BWG!!!! The figure of FI is not shown!!!

Corrected

---

## [Decision Letter · Decision Letter 1]

8 Nov 2021

Evaluation of dietary crude protein concentrations, fishmeal, and sorghum inclusions in broiler chickens offered wheat-based diet via Box-Behnken response surface design

PONE-D-21-26429R1

Dear Dr. Liu,

We’re pleased to inform you that your manuscript has been judged scientifically suitable for publication and will be formally accepted for publication once it meets all outstanding technical requirements.

Kind regards,

Arda Yildirim, Ph.D.

Academic Editor

PLOS ONE

Additional Editor Comments (optional):

Thanks for sincerely and thoroughly considering and attending to the comments and concerns.

Reviewers' comments:

Reviewer's Responses to Questions

**Comments to the Author**

1. If the authors have adequately addressed your comments raised in a previous round of review and you feel that this manuscript is now acceptable for publication, you may indicate that here to bypass the “Comments to the Author” section, enter your conflict of interest statement in the “Confidential to Editor” section, and submit your "Accept" recommendation.

Reviewer #2: All comments have been addressed

Reviewer #3: All comments have been addressed

2. Is the manuscript technically sound, and do the data support the conclusions?

Reviewer #2: Yes

Reviewer #3: Yes

3. Has the statistical analysis been performed appropriately and rigorously? 

Reviewer #2: Yes

Reviewer #3: Yes

4. Have the authors made all data underlying the findings in their manuscript fully available?

Reviewer #2: Yes

Reviewer #3: Yes

5. Is the manuscript presented in an intelligible fashion and written in standard English?

Reviewer #2: Yes

Reviewer #3: Yes

6. Review Comments to the Author

Reviewer #2: The authors response is satisfactory. The revised manuscript is acceptable for publication. The formatting editor could check the resolution of figure to be included in the published version.

Reviewer #3: (No Response)

7. PLOS authors have the option to publish the peer review history of their article (what does this mean?). If published, this will include your full peer review and any attached files.

Reviewer #2: **Yes: **Amit Kumar Singh

Reviewer #3: No

---

## [Editor Report · Acceptance letter]

10 Nov 2021

PONE-D-21-26429R1 

Evaluation of dietary crude protein concentrations, fishmeal, and sorghum inclusions in broiler chickens offered wheat-based diet via Box-Behnken response surface design 

Dear Dr. Liu:

I'm pleased to inform you that your manuscript has been deemed suitable for publication in PLOS ONE. Congratulations! Your manuscript is now with our production department. 

Kind regards, 

on behalf of

Prof. Dr. Arda Yildirim 

Academic Editor

PLOS ONE